# Mutant fixation in the presence of a natural enemy

Dominik Wodarz [1,2,3] ✉ & Natalia L. Komarova[2,4]

The literature about mutant invasion and fixation typically assumes populations to exist in isolation from their ecosystem. Yet, populations are part of ecological communities, and enemy-victim (e.g. predator-prey or pathogen-host) interactions are particularly common. We use spatially explicit, computational pathogen-host models (with wild-type and mutant hosts) to re-visit the established theory about mutant fixation, where the pathogen equally attacks both wild-type and mutant individuals. Mutant fitness is assumed to be unrelated to infection. We find that pathogen presence substantially weakens selection, increasing the fixation probability of disadvantageous mutants and decreasing it for advantageous mutants. The magnitude of the effect rises with the infection rate. This occurs because infection induces spatial structures, where mutant and wild-type individuals are mostly spatially separated. Thus, instead of mutant and wild-type individuals competing with each other, it is mutant and wild-type "patches" that compete, resulting in smaller fitness differences and weakened selection. This implies that the deleterious mutant burden in natural populations might be higher than expected from traditional theory.

In reproducing populations, evolution is driven by the generation of new mutations, and the fate of the mutants is determined by selection and drift. The dynamics of mutant invasion have been studied extensively in a variety of settings[1,2], driven in large part by the analysis of mathematical models. The fixation probability of a mutant is a central concept in this respect[1,3,4]. It is defined as the probability for a mutant that has been introduced into a population to rise and replace the wild-type. The conditional average time to fixation of a mutant is another important measure, determined across those instances where mutant fixation occurs. An extensive literature exists assuming constant populations[3,5–7], which can be described mathematically by e.g. the Moran process or the Fisher-Wright process. Much of this work assumes well-mixed populations, although important insights into the dynamics of mutant invasion have been obtained for spatially or deme-structured populations[8–13], as well as more generally for mutant fixation on graphs[14–18]. Besides constant populations, the effect of demographic fluctuations

around an equilibrium on the probability of mutant fixation has also been analyzed[19–24].

Evolutionary theory about mutant fixation has typically focused on the evolving population in isolation, which has given rise to many fundamental insights. In nature, however, evolving populations exist as part of an ecosystem. Natural enemies present a particularly common ecological setting. Yet, it is currently unclear how the presence of a natural enemy (that equally attacks both wild-type and mutant individuals) influences the fixation probability of a mutant. Within such a system, the evolving population can persist around an equilibrium, which at first sight seems similar to a constant population scenario. In spatially structured (and hence biologically realistic) models, however, the stable persistence of the population can be the result of continuous local extinction events coupled with migration of individuals into temporary refuge spaces without enemies, as illustrated by patch and metapopulation models[25]. Population fluctuation, frequent extinction events, and bottlenecks have been shown to change the properties of

[1]Department of Population Health and Disease Prevention, University of California, Irvine, CA 92697, US. [2]Department of Mathematics, University of California, Irvine, CA 92697, US. [3]Present address: School of Biological Sciences, Ecology, Behavior & Evolution Department, University of California San Diego, 9500 Gilman Drive, La Jolla, CA 92093, USA. [4]Present address: Department of Mathematics, University of California San Diego, 9500 Gilman Drive, La Jolla, CA 92093, USA. ✉ e-mail: dwodarz@ucsd.edu

mutant invasion[20,26,27], and hence it is important to study the spatial dynamics of mutant invasion in the presence of a natural enemy.

Here, we study the properties of mutant invasion and fixation in spatially structured populations at quasi-equilibrium, assuming that the evolving population exists in the presence of a natural enemy. While applicable to all enemy-victim settings, the model is formulated as a population of cells that are subject to infection by a virus (regardless of cell genotype). We start by considering a spatial stochastic agent-based model and then compare its properties to those of patch models.

## Results

### Spatial agent-based model of host evolution in the presence of infection

We consider an agent-based model (ABM) on a 2D grid, where each of $n_1 \times n_2$ spots could be either empty or contain an uninfected or infected cell of different types (wild-type or mutant). Each time-step consists of $N_I$ elementary updates, where $N_I$ is the number of currently occupied sites. At each elementary update, a random cell is picked, and the following events can occur:

- If an uninfected cell is picked it can attempt division with a probability $R$. A random spot among the 8 nearest neighbors is chosen, and if unoccupied, the offspring cell is placed there.
- With a probability $D$, the uninfected cell dies.
- With a probability $1-R-D$, no event occurs for the uninfected cell.
- Infected cells are assumed not to divide. When selected, they can die with a probability $A$.
- Infected cells attempt an infection event with a probability $B$, during which a random spot is chosen among the 8 nearest

neighbors. The infection proceeds if the chosen spot contains an uninfected cell.

- With probability $1-A-B$, no event occurs for infected cells.

Periodic boundary conditions were used in all simulations.

In the absence of mutants, i.e. just one cell type in a spatial pathogen-host system, the dynamics have been well defined. Over time, the population sizes of uninfected and infected cells converge to a state where they fluctuate around an equilibrium (Fig. 1); we will refer to the mean equilibrium size of the uninfected population as $N_u$. The spatial distribution of cells, however, strongly depends on the rate of infection. For relatively low infection rates, the cells are distributed more uniformly through space (Fig. 1A). For higher infection rates, however, pronounced spatial structures emerge in which moving fronts of uninfected cells are "chased" by infected cells (Fig. 1B). In a particular local area, the infection drives the cell population extinct. Spatial separation of uninfected cells from sources of infection, due to cell divisions to adjacent spots, leads to the persistence of the populations on a global level (across the entire grid). This recapitulates the well-known spatial refuge effect that can contribute to population persistence and spatial pattern formation in predator-prey dynamics[25].

To study the fixation probability and conditional fixation times of mutant cells, we start with a square block of uninfected wild-type individuals in the middle of the grid and place a smaller square block of infected wild-type individuals inside this block. We then simulate population dynamics in the absence of mutants for a certain amount of time, until quasi-equilibrium is reached (the exact choice of the initial condition is unimportant for reaching this state). Next, we replace one (or several) randomly selected uninfected wild-type cells with a

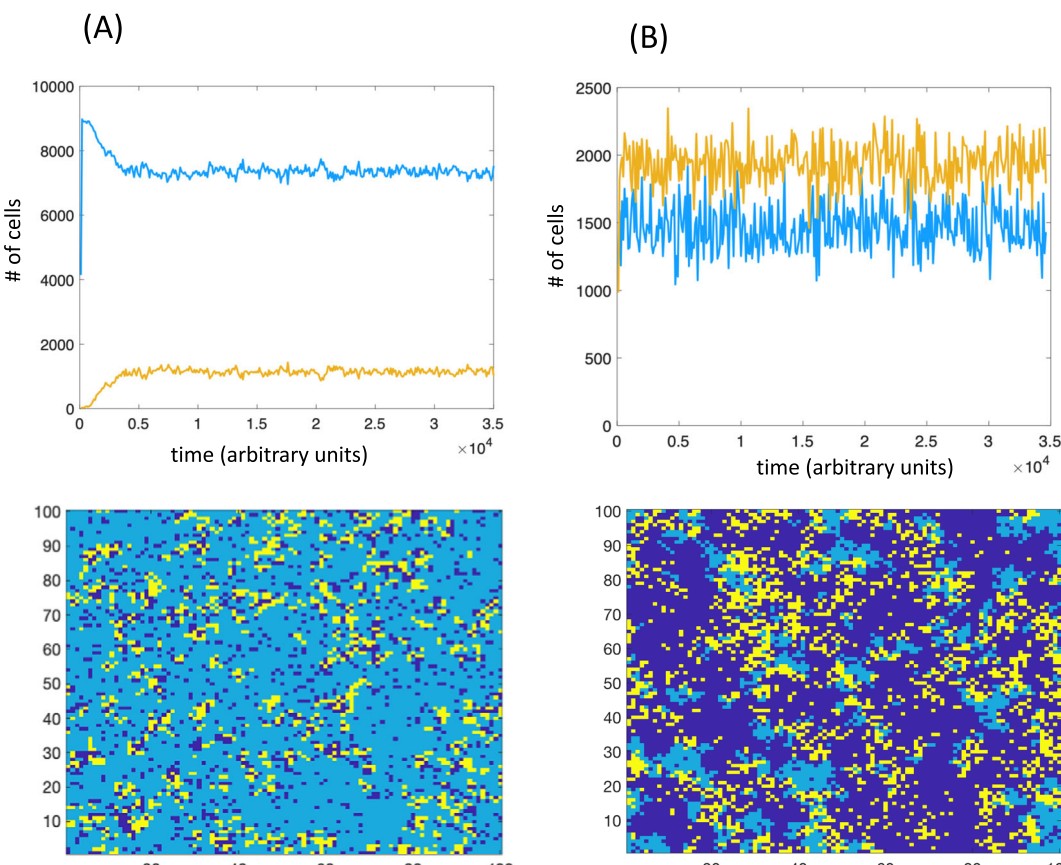

**Fig. 1 | Basic dynamics of infected and uninfected cells without evolution.** For the time series, light blue and yellow colors represent the populations of uninfected and infected cells. For the spatial pictures, light blue and yellow colors also represent uninfected and infected cells. Dark blue shows empty space. **A** Low infection rate. **B** High infection rate. $R = 0.5$; $D = 0.05$; $A = 0.1$; $n_1 = n_2 = 100$. For panel (**A**) $B = 0.2$; for panel (**B**) $B = 0.9$.

corresponding number of mutant cells, at quasi-equilibrium. This initial mutant placement occurs at a moment in the simulation when the number of uninfected wild-type cells equals the equilibrium value, $N_u$, determined numerically by calculating the long-term temporal average of the population size. If several instead of one mutant cell are introduced, multiple randomly chosen wild-type uninfected cells are turned into mutants at the same time. No de novo mutations are considered. We then allow the simulation to run until either the mutant is extinct, or until the mutant population has replaced the wild-type cell population (mutant fixation). The simulation is run repeatedly and the fraction of runs during which mutant fixation occurs is determined. For the cases of mutant fixation, the time until fixation is determined (conditional fixation time).

This setup unites evolutionary theory on graphs[14–18] with pathogen-host or predator-prey dynamics. As a consequence, we are not studying mutant evolution in a constant population, but in a population that changes dynamically (both temporally and spatially), driven by the pathogen-host interactions. Thus, before mutant introduction, uninfected and infected populations fluctuate around a steady state, with the fluctuations being more pronounced for higher infection rates. Mutant uninfected cells are introduced into this system, and the evolutionary fate of the mutant uninfected cells is followed. Mutant infected cells do not contribute to mutant spread in this model, because infected cells are assumed not to divide, and to die after a certain period of time.

These dynamics are investigated for neutral, advantageous, and disadvantageous mutants, comparing the results in the absence and presence of infection, where the rate of infection is varied. To implement fitness advantage/disadvantage of mutants, we assume that the division rate of cells is increased/decreased relative to the wild-type by multiplying it with the coefficient $(1 + s)$, where $s$ is the selection coefficient. An advantageous mutant corresponds to $s > 0$, and for a disadvantageous mutant $s < 0$.

When comparing the dynamics under different infection rates, equilibrium population sizes vary. To control for this, the grid size is adjusted such that the average equilibrium population size of uninfected cells, $N_u$, is kept approximately constant, regardless of infection rate. Probabilities of mutant fixation in the presence of infection are compared with those in the absence of infection, and also with the well-known formula for the Moran process of equivalent size, $P_{fix}(i) = \frac{1-1/(1+s)^i}{1-1/(1+s)^n}$, where $i$ denotes the initial number of mutants in a total population of $n$ individuals ($N_u$ in our setting).

For neutral mutants, as expected, the numerically obtained fixation probability is $1/N_u$ regardless of infection rate. This is identical to the fixation probability provided by the constant population Moran process, with the population size given by the number of uninfected cells. Only the population of uninfected cells determines the fixation probability because the infected cells are assumed to not divide in our model. Therefore, once infected, the cells are destined to die in time, determined by the death rate of infected cells, A.

For advantageous mutants, we find that the presence of infection weakens selection (Fig. 2A). Without infection, the numerically obtained mutant fixation probability is very close to the value predicted by the non-spatial Moran process (for a discussion of the role of demographic fluctuations see ref. 20); therefore, here and in other cases below, the Moran formula is a convenient reference point for evaluating the changes in fixation probability due to infection. In the presence of the virus, the fixation probability is noticeably reduced, with larger reductions observed for faster infection rates. The average conditional time to fixation is found to be lower in the presence compared to the absence of infection, with shorter fixation times occurring for faster infection rates.

For disadvantageous mutants, selection is again found to be weakened in the presence of the virus (Fig. 2B). Without infection, the numerically obtained mutant fixation probability is again close to the value predicted by the non-spatial Moran process. In the presence of infection, however, the fixation probability of disadvantageous mutants is noticeably increased, with the larger increases seen for faster infection rates. An up to 1000-fold increase in the fixation probability is seen for the parameter regime studied here (Fig. 2B). The average conditional time to fixation is again shortened by the virus (Fig. 2B).

## Deme models of host mutant evolution in the presence of infection

An alternative and coarser-grained method of modeling spatial interactions are deme or patch models. Rather than tracking each individual and their spatial location, populations in the demes are assumed to be well-mixed. In addition, individuals migrate between patches, and migration can be spatially restricted to nearest neighbors, or less spatially restricted. Here we consider a two-dimensional deme/patch model, consisting of $n_1 x n_2 = \mathcal{N}$ patches. In each patch, host-pathogen dynamics are described by stochastic Gillespie-type simulations of ODEs, given by

$$\frac{dx_i}{dt} = rx_i\left(1 - \frac{x_i + z_i}{K}\right) - dx_i - \beta x_i z_i + \mu\left(\frac{\sum_k x_k}{\mathcal{N}} - x_i\right), \quad (1)$$

$$\frac{dz_i}{dt} = \beta x_i z_i - az_i + \mu\left(\frac{\sum_k z_k}{\mathcal{N}} - z_i\right), \quad (2)$$

where $x_i$ and $z_i$ in Eqs. (1 and 2) denote the populations of susceptible and infected cells in patch $i$. Lower case letters are used for rate parameters, as opposed to the capital letters used for probabilities in the agent-based model: the parameter $r$ is the basic division rate of cells, $d$ is the death rate of susceptible cells, $\beta$ is the rate of infection, and $a$ stands for the death rate of infected cells. Migration of uninfected and infected cells to/from other patches occurs with a rate $\mu$, and migration can occur either to/from any patch in the system (non-spatial migration, shown in the equations above), or to/from the eight nearest neighbors (spatial migration, see Supplementary Note 1, Section 1.1). $K$ is the carrying capacity of a patch. For stochastic simulations of mutant spread, the Gillespie-type algorithm is applied to this patch ODE model with details given in the Supplementary Note 1 (Section 1.1). When varying the infection rate of the virus, the grid size is again adjusted to maintain approximately the same total number of uninfected cells across the different infection rates.

Over time, the sum of populations in this model fluctuates around a steady state value (Fig. 3). Although there is a global quasi-steady state, within each patch, populations can crash to extinction due to the infection, and can subsequently be re-colonized to repeat this pattern[25]. To investigate mutant invasion, we introduced the mutants when the total global population size of uninfected cells is equal to the rounded temporal average of the global population, $N_u$. To introduce a mutant, a random patch was selected with the probability proportional to its number of uninfected wild-type individuals, and in that patch, a single uninfected wild-type individual was replaced by a mutant one. To introduce multiple mutants, this procedure was repeated as many times as the desired number of mutants. The fixation probabilities and times were determined in the same way as for the ABM.

We observe results that are qualitatively similar to the ABM described above. That is, selection is weakened both for advantageous and disadvantageous mutants (Fig. 4): the fixation probability of advantageous mutants decreases below the value predicted by the non-spatial Moran process as the infection rate rises, and the fixation probability of disadvantageous mutants increases above that of the Moran process. The conditional time to fixation is generally again reduced by the presence of the infection (Fig. 4), although for spatial migration the dependence can be non-monotonous. In previous work, conditional fixation times have been shown to follow complex patterns[28], and it is beyond the scope of the current study to investigate this in further detail. It is interesting to note that these trends

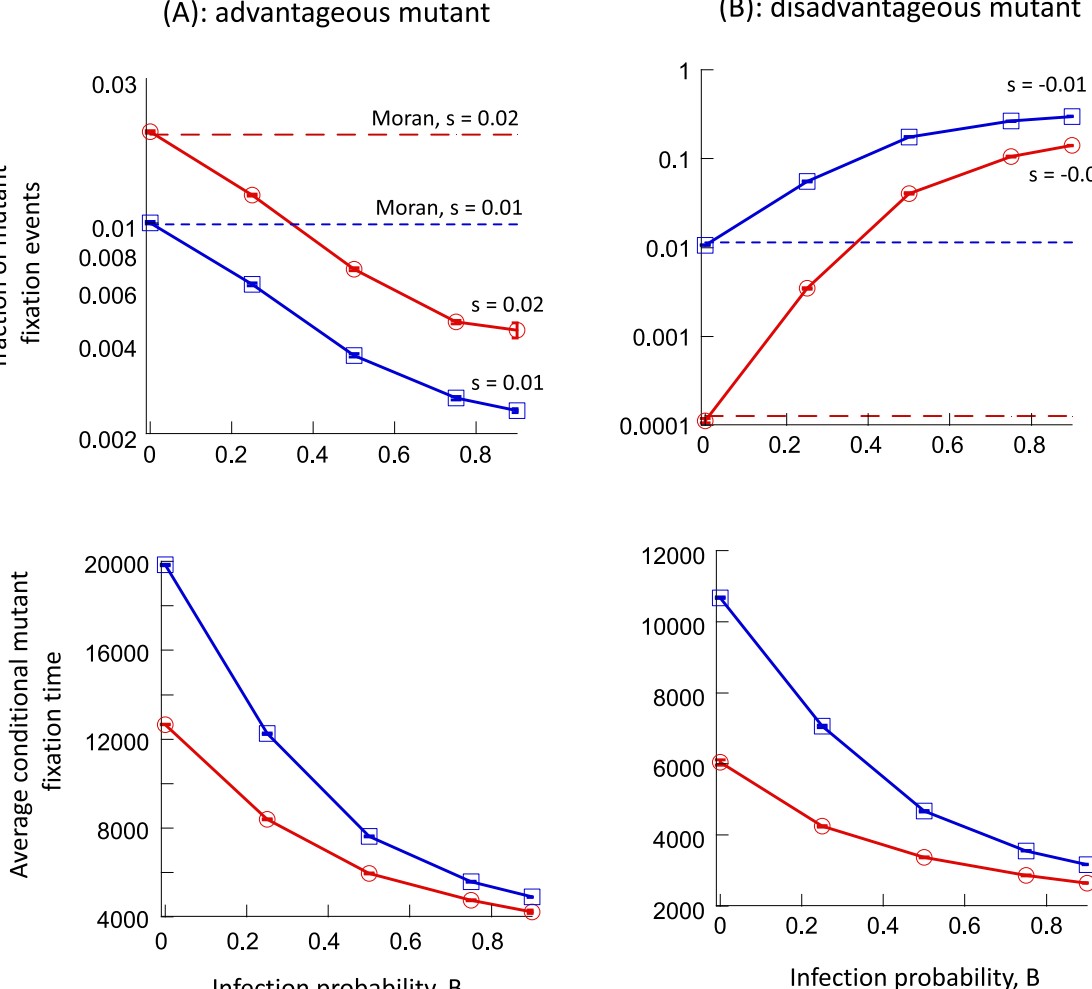

**Fig. 2 | Fixation probabilities (with 95% confidence limits) and conditional fixation times (with standard errors) in the agent-based model for different infection probabilities. A** Advantageous mutants for two different values of selection coefficient, s. 1 mutant was introduced. **B** Disadvantageous mutants for two different values of selection coefficient, s. 500 mutants were introduced. Parameters are as follows. $R = 0.5$; $D = 0.05$; $A = 0.1$; $N_u = 944$. The grid sizes for the successive infection probabilities are: $32 \times 33$, $42 \times 42$, $60 \times 60$, $73 \times 73$, $81 \times 80$.

apply both to simulations with spatial and non-spatial migration (Fig. 4).

**Patch versus cell competition**

We propose that the reason for the weakened selection observed in the presence of infection is the behavior of the system as a metapopulation, regardless of the underlying model. That is, cells go extinct locally as a result of the infection, and persist by colonizing other areas of space, which temporarily do not contain infection and hence provide a refuge for the cells. This happens across a continuous space in the ABM, as shown in Fig. 1. It happens more explicitly in the patch models where patches periodically go extinct and become recolonized (Fig. 3), both under spatial and non-spatial migration. For relatively large infection rates, this also leads to a spatial separation of wild-type and mutant cells. In terms of the patch model, a patch is likely to either contain only wild-type cells or only mutant cells, but rarely both. In this setting, mutant and wild-type patches (rather than cells) effectively compete for colonization of empty patches, and this leads to mutant fixation probabilities that deviate noticeably from those predicted by the Moran model (or the process without infection). For low infection rates or in the absence of infection, mixing of wild-type and mutant cells is more likely, and it is the competition among cells (rather than patches) that drives

mutant fixation. Consequently, the observed fixation probabilities converge to those predicted by the Moran process. For intermediate infection rates, the fixation probabilities are determined by a mixture of cell and patch competition.

We demonstrate this in more detail using the patch model with non-spatial migration, see Fig. 5. Assuming that mutant and wild-type cells do not co-occur in the same patches (panel (a)), we can write down a coarse-grained model where patches, rather than individual cells, are agents, and where population dynamics are governed by the following processes (panel (b)): empty patch colonization, patch infection, infected patch extinction, and patch conversion (a cell of the other type migrating into an uninfected patch and taking over). Let us denote by $X, Y,$ and $Z$ the total numbers of uninfected wild-type patches, uninfected mutant patches, and infected patches, and by $w_x, w_y,$ and $w_z$ the mean per-deme populations of these types of cells, in patches of types $X, Y,$ and $Z$ respectively. We can summarize the coarse-grained dynamics as follows:

$$\dot{X} = \mu X w_x \left(1 - \frac{X+Y+Z}{\mathcal{N}}\right) P_{col}^x - \mu Z w_z \frac{X}{\mathcal{N}} P_{inf}^x - \mu Y w_y \frac{X}{\mathcal{N}} P_{x \to y}$$
$$+ \mu X w_x \frac{Y}{\mathcal{N}} P_{y \to x}, \tag{3}$$

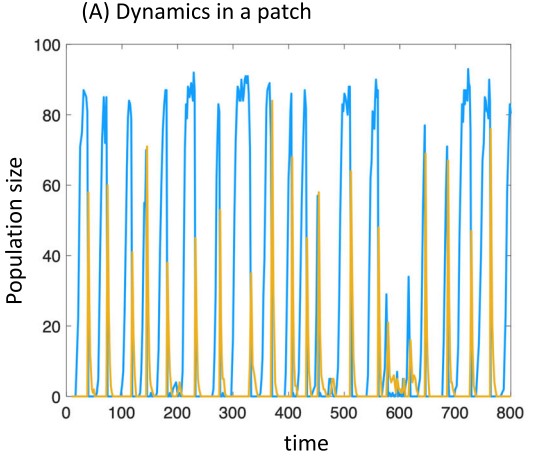

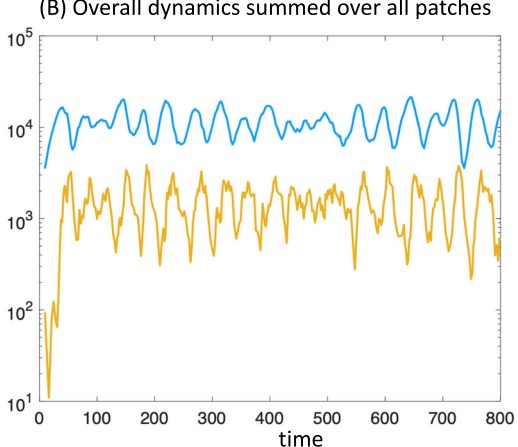

**Fig. 3 | Dynamics in the patch model with migration to nearest neighboring patches.** Blue and orange colors represent the populations of uninfected and infected cells. **A** Dynamics within a patch. **B** Total dynamics, with population sizes summed up across all patches. Parameters are as follows: $r = 0.7$; $d = 0.1$; $\beta = 0.5$; $a = 0.5$; $K = 100$, $\mu = 0.02$, $n_1 = n_2 = 19$.

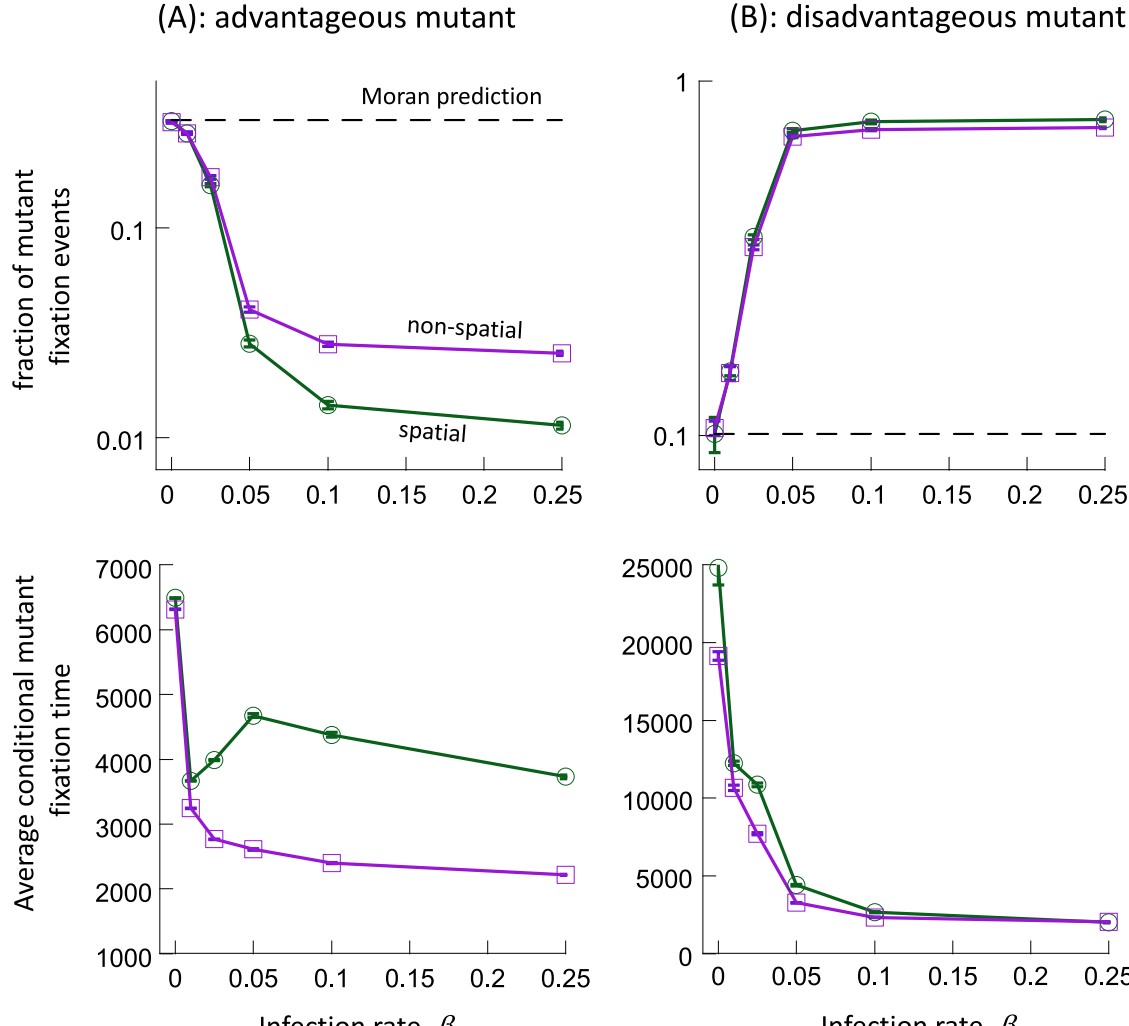

**Fig. 4 | Fixation probabilities (with 95% confidence limits) and conditional fixation times (with standard errors) in the deme model with spatial and non-spatial migration. A** Advantageous mutants, $s = 0.02$; 20 mutants were introduced. **B** Disadvantageous mutants, $s = -0.001$; 9000 mutants were introduced. Parameters were as follows: $r = 0.7$; $d = 0.1$; $a = 0.5$; $K = 100$, $\mu = 0.02$; $N_u = 11{,}288$. For spatial migration, the grid sizes for the successive infection rates are $11 \times 12$, $14 \times 14$, $22 \times 21$, $22 \times 22$, $20 \times 21$, $20 \times 19$. For non-spatial migration, they are $11 \times 12$, $14 \times 14$, $22 \times 22$, $24 \times 24$, $23 \times 24$, $22 \times 23$.

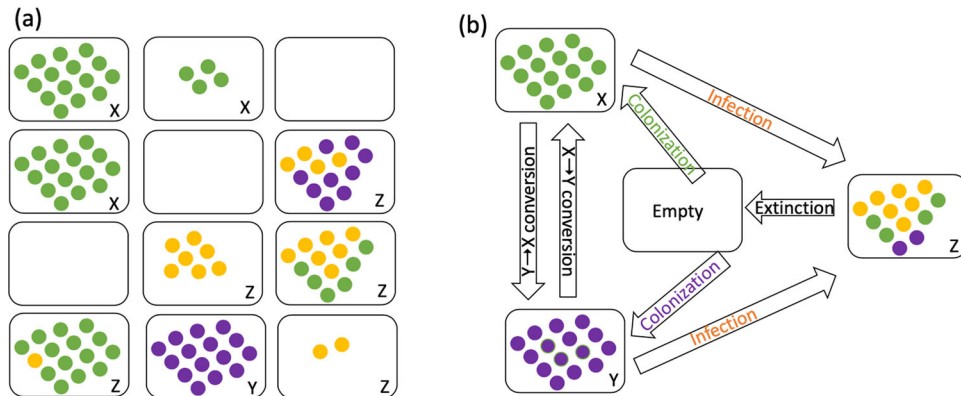

**Fig. 5 | A schematic explaining the components of mutant dynamics in patches.**
**a** Populations of patches can be uninfected wild-type ($X$), uninfected mutant ($Y$), and infected ($Z$). **b** Coarse-grained model structure includes processes of colonization, infection, extinction, and conversion. Green and purple dots denote uninfected wild-type and mutant cells, and red dots indicate infected cells.

$$\dot{Y} = \mu Y w_y \left(1 - \frac{X + Y + Z}{\mathcal{N}}\right) P_{col}^y - \mu Z w_z \frac{Y}{\mathcal{N}} P_{inf}^y + \mu Y w_y \frac{X}{\mathcal{N}} P_{x \to y}$$
$$- \mu X w_x \frac{Y}{\mathcal{N}} P_{y \to x}, \tag{4}$$

$$\dot{Z} = \mu Z w_z \frac{X P_{inf}^x + Y P_{inf}^y}{\mathcal{N}} - \frac{Z}{T_{ext}} \tag{5}$$

The first term in the right hand side of the Eqs. (3) and (4) describes colonization of empty patches, the second term describes the process of infection (the same terms reappear in Eq. (5) with the opposite signs). The remaining terms in Eqs. (3) and (4) describe conversion, and the last term in Eq. (5) is patch extinction (see Supplementary Note 2 for complete details of this model). Denote the equilibrium number of uninfected patches, in the absence of mutants, by $X_{eq}$. At the level of patch competition, there are two selection coefficients: one associated with patch conversion ($s_p^{conv}$) and the other with the process of extinction-recolonization ($s_p^{ext-rec}$). The overall selection coefficient is given by the mean of these two, weighted with their respective rates, $R_{conv}$ and $R_{ext-rec}$:

$s_p = \frac{R_{ext-rec} s_p^{ext-rec} + R_{conv} s_p^{conv}}{R_{ext-rec} + R_{conv}}$, and the probability of mutant demes taking over can be calculated by the Moran formula, applied at the level of patches: $\rho_i = \frac{1 - 1/(1 + s_p)^i}{1 - 1/(1 + s_p)^{X_{eq}}}$, where $i$ is the initial number of mutant patches. If the majority of demes are occupied and conversion is the main driving force of mutant dynamics, then the selection coefficient is approximately $s_p^{conv}$, and the probability of mutant takeover is approximately that predicted by the usual Moran process in the absence of infection (and with the equivalent population size). This is because the process of deme conversion through death-birth events is the same as what comprises the conventional dynamics that leads to the Moran formula. If, however, the system of patches is sparsely populated and the process of recolonization is the leading force of mutant spread, then we have $s_p \approx s_p^{ext-rec}$, where $s_p^{ext-rec} = \frac{w_y P_{col}^y P_{inf}^x}{w_x P_{col}^x P_{inf}^y} - 1$, containing quantities $P_{col}^x = 1 - d_x/r_x$ and $P_{col}^y = 1 - d_y/r_y$, which are the probabilities for a single wild-type (mutant) cell to successfully colonize a patch, and $P_{inf}^x = 1 - \left(\frac{w_x \beta}{a}\right)^{-1}$ and $P_{inf}^y = 1 - \left(\frac{w_y \beta}{a}\right)^{-1}$, which are the probabilities for a single infected cell to start successful infection in a wild-type (mutant) patch. The selection coefficient corresponding to

the process of recolonization has a much smaller absolute value compared to $s_p^{conv}$, and describes much weaker selection. In the Supplementary Note 2, we provide a comparison of the prediction of the coarse-grained model with numerical simulations of the full system, and find them in close agreement (in the region of model applicability). While the coarse-grained approach is somewhat limited in that it does not describe patch models with spatial migration, or spatially distributed ABM systems, it provides a way to separate the different processes that contribute to mutant dynamics, and to explain the observation of a significant selection reduction in systems with infection.

## Discussion

We have shown that the dynamics of mutant invasion can be strongly altered if both wild-type and mutant individuals are equally susceptible to a natural enemy such as an infection. In particular, we showed that this can lead to a significant weakening of selection. Thus, the fixation probability of a disadvantageous mutant can be strongly increased compared to the evolutionary dynamics in the absence of infection, and the fixation probability of advantageous mutants can be substantially lowered. The difference can be several orders of magnitude. In the presence of infection, it is possible that a deleterious mutant, whose invasion potential can be neglected according to traditional theory, has a reasonable chance of emerging. These results are highly relevant for natural settings, because most populations do not exist in isolation, but are part of a larger ecosystem that includes natural enemies.

In this analysis, we considered the simplest infection system, which corresponds most closely to other natural enemy interactions, such as predator-prey or parasitoid-host interactions. Once infected, an individual is destined to die. The infected individual is assumed not to reproduce anymore, or to recover from the infection. Hence, this would represent an SI (susceptible-infected) system. In future work, it would be interesting to extend our framework to more complex infection systems, such as SIR (susceptible-infected-recovered), or SIRS models, or to models in which infected individuals are assumed to reproduce, thereby transmitting the pathogen vertically.

We only analyzed spatially explicit models in this study, because spatial dynamics are well-known to lead to more realistic and stable enemy-victim dynamics, and lack of any spatial population structure typically results in highly oscillatory dynamics that are likely to lead to population extinction in stochastic models. Using a combination of numerical and analytic methods, we have shown that the mechanism underlying the weakened selection in this setting is a result of the population structure assumed in our models. In our system, an

increase in the infection rate results in dynamics that are progressively less stable on a local level. That is, local population extinction occurs frequently, and persistence across space is possible because host populations keep dividing or migrating (in the deme model) into new locations where they can grow temporarily without the infection present[25]. In this setting, it is not the competition between individuals that matters because a given location rarely contains both mutant and wild-type individuals together. Instead, most locations contain either wild-type or mutants, and patches compete with each other in the context of extinction and recolonization of local areas/patches/demes. This leads to a much lower difference in fitness of mutant compared to wild-type individuals, accounting for the large differences seen in the fixation probabilities. These dynamics are related to extinction-recolonization dynamics that have been explored in the absence of natural enemies, also demonstrating weakened selection[26].

It is not possible to directly compare the evolutionary dynamics in the spatial systems analyzed here to a corresponding non-spatial setting. For an equivalent parameter set, the dynamics in the absence of spatial population structure result in population extinction. For other parameter regions, where the system persists in the absence of spatial structure, it is possible to investigate how infection influences the fixation probability of the mutant in a non-spatial system. In this case, the infection introduces an increase in demographic fluctuations around equilibrium. Previous work has shown that the existence of demographic fluctuations can independently result in weakened selection (compared to a constant population Moran process)[20]. However, this is a different mechanism, and based on preliminary simulations, selection is only weakened modestly compared to the magnitude of the effect observed in our spatial dynamics. The reason is that population persistence in the absence of spatial structure is only possible for relatively slow infection rates, which only introduce a limited increase in demographic fluctuations. A more detailed and comprehensive examination of the effect of an infection on demographic fluctuations and mutant fixation in non-spatial systems is subject to future work.

Experimental data that could be used to address the theoretical predictions presented here are so far not available. Bacteria-phage interactions[29] are a biological system in which these dynamics could be explored. In other contexts, bacteria and phages have been used to study aspects of virus-host evolution experimentally. Phage infections have been shown to have a direct impact on the evolution of the bacterial cell populations[30–33]. For example, the presence of bacteriophages can increase the evolutionary potential of the bacterial population, which can provide a benefit to the bacteria that balances the increased cell mortality induced by the infection[30]. In the context of antibiotic resistance evolution in bacteria, phage infections have been shown to result in the emergence of bacterial strains that are resistant to the phage, and interestingly, phage-resistance could lead to the simultaneous increase in bacterial sensitivity to the antibiotic[34,35]. Other examples of the eco-evolutionary dynamics of phages and bacteria are discussed in references[29,36,37].

Our present investigation has focused on very basic evolutionary dynamics, i.e. revisiting the theory on mutant invasion if the population is subject to infection, rather than evolving in isolation. The evolutionary processes considered in our models are not in response to the selection pressure exerted by the infection, but represent infection-unrelated evolutionary changes that can be disadvantageous, advantageous, or neutral. This is in contrast to an extensive literature about host-pathogen co-evolution[38] that typically investigates scenarios where the infection is a driver of (co-)evolutionary change. The work presented here, however, has shown that the presence of an infection, or a natural enemy in general, can fundamentally alter the basic dynamics of mutant invasion in a population, through spatial heterogeneity generated by underlying dynamics.

## Methods

Computational models are used to analyze the dynamics of mutant fixation in the presence of a natural enemy (pathogen) that equally attacks wild-type and mutant individuals. The models are spatially explicit, given by (i) an agent-based model that tracks individuals and their spatial location (see detailed description in conjunction with Fig. 1 above), and by (ii) a patch or deme model, tracking the population dynamics within demes that are connected to each other by migration. The latter is presented by ordinary differential equations of host-pathogen dynamics (see basic Eqs. (1)–(2) and Supplementary Note 2, Section 2.1 for the equations in the presence of two types of hosts). Stochastic simulations were used to sample trajectories from the mean field description, which are described in Supplementary Note 1, Section 1.1. Further, a coarse-grained approximation of these dynamics was developed both in the absence (Supplementary Note 1) and in the presence (Supplementary Note 2) of mutants. This method allows studying mutant dynamics in a spatially distributed system at the level of patches, and it provides tools to disentangle different mechanisms that contribute to patch selection (Supplementary Note 2, Section 2.3). The excellent agreement of this approximation with stochastic simulations is demonstrated in Supplementary Note 2, Section 2.4.

### Reporting summary

Further information on research design is available in the Nature Portfolio Reporting Summary linked to this article.

## Data availability

The results reported in this manuscript are based on mathematical analysis and corresponding computer simulations (and not empirical data). The mathematical analysis is fully described in the main text and the Supplementary Information. Source Data used to generate figures can be found within Supplementary Code 1, Supplementary Code 2, and Supplementary Code 3.

## Code availability

Computer codes for the agent-based model and the patch models are provided as Supplementary Code 1, Supplementary Code 2, and Supplementary Code 3.

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

## Acknowledgements
The work was funded by NSF grant DMS 2152155 (N.L.K. and D.W.)

## Author contributions
D.W. and N.L.K. designed the study, performed analysis, and wrote the paper.

## Competing interests
The authors declare no competing interests.
