## [Peer Review File · Nature Communications]

Mutant fixation in the presence of a natural enemyReviewers' Comments:

Reviewer #1:

Remarks to the Author:

In this paper, the authors study how the presence of a natural enemy, here infection, would influence the fixation probability and fixation time of mutants.

It is shown that the presence of infection weakens selection and the time to fixation decreases both in the spatial and non-spatial migration. They argue that this happens because the infection causes the population to be spatially separated. If the rate of infection is high enough the mutants and wild-types do not cooccur in the same patch and as a result it is not mutant and wild-type individuals that compete with each other but rather the mutant and wild-type patches.

They have studied this using a coarse-grained ODE model and also simulation.

I think this paper is well-written and novel enough and is totally capable of being published in the nature communication journal. However, I have some questions and I would appreciate it if the authors make them clear to me.

1) In the section 2.2 line, I don't understand what you mean by stochastic Gillespie of the appropriate ODE. Do you use the parameters obtained from ODE in the simulation? Or does it mean the dynamics within the patches deterministic but the migration or the dynamics between the patches is stochastic?

2) In Eqs. 25 and 26 of supplementary information I guess what should contribute to the fitness of the mutant should be the success of the mutant and not its failure. Shouldn't it be for example:

$w_y^{(Y)} * P_{x \rightarrow y}$ instead of $w_x^{(X)} * P_{y \rightarrow x}$?

3) In the Fig. S4 I wonder what is different between mutants and wild-types. Apparently, they both have the same death and reproduction rates.

4) In the Fig. S4 does each frame constitute of 10,000 updating steps? If so, why did you make this setting?

5) How do you calculate the conversion rate $P_{x \rightarrow y}$ or $P_{y \rightarrow x}$? I didn't see anywhere in the paper mentioning this.

And finally, two minor comments:

1) In the supplementary information section 2.2 equations 20-21 parameter Z does not have a subscript in some cases.

2) In the equation after Eq. 26 I think the subscript of s should be p. Am I right?

Reviewer #2:

Remarks to the Author:

The manuscript connects two very different fields: The very abstract field of evolutionary graph theory (with the probability of fixation as a central object) and ecological interactions in the form of predator-prey or pathogen host dynamics.

In this kind of ecological setting, a lot of people will think of an internal equilibrium between U and I. The catch here is that there are two types of U, wild type and mutant. It took me some time to grasp this, it is a very nice approach, but I am not sure if this could not be spelled out more clearly. In addition, the model is very different from the usual fixation probability considerations on graphs, as the population size varies – this would correspond to a dynamically changing graph, which is challenging. Thus, it is very nice that the authors keep the spatial structure relatively simple.

I think this is a great piece of work which opens the door for both associated fields to widen their scope. I would have many ideas how to build on this and I would be happy to see this paper published in a prominent place.

However, I have a number of issues with this model (actually, they all seem minor to me) that the authors could consider in a revision:

- Would it make sense to show also the associated mean field theory? From simulations, I see

fluctuations in the population sizes, but I was wondering if I am correct in translating this into a predator-prey system with a carrying capacity for the prey, which would have a stable internal fixed point? Can you calculate an approximation for N_u from this?

- Can I think about the dynamics between uninfected, infected and empty space as something resembling an SIRS model? Maybe that analogy would work for the infection case and help some readers to grasp what happens here.

- The role of infected cells in the mutation process is not entirely clear to me. If I understand correctly, mutations can only occur in uninfected (=dividing) cells. Can I consider the infected cells as agents that effectively increase the death rate of their neighbors, but otherwise not relevant? Is that the reason to discard infected cells from the population size when comparing fixation probabilities? But if there are many infected cells, the population is prone to decline fast – so their presence cannot fully be discarded. Is that another reason the selection appears weaker with infection?

- I found section 2.1 under the heading “results” a bit awkward. I would have preferred to have this as its own section on the model as such, but I understand that this is sometimes not possible due to unlucky formatting requirements driven entirely by the needs of experimental biology.

- In section 2.1, a bit more structure could help, e.g. have bullet points for the different events.

- I also found the notation a bit unlucky, maybe the probabilities can have more intuitive names? Is it the idea that the probabilities in the agent based models have capital letters (e.g. “A”) and the associated rates in the ODE small letters (e.g. “a”)? If so, please write that.

- On p.4, you write about “continuous movement of cells”, but according to the basic model descriptions, cells do not move. Please clarify!

- When you introduce several mutants, how do you place them? I assume that this makes a major difference.

- “Significantly reduced” – I am sure you can make it significant by simulating more, but I suggest to drop this word as some people think p-values when they see significant.

- “stochastic Gillespie simulations of the appropriate ODEs” is very complex. I assume that you take the rates of an individual based process and use them either to derive a deterministic ODE or to do Gillespie-type simulations. In principle, there are many individual based rate processes that lead to a single ODE, so I do not think one can just say that we work on the corresponding stochastic model. The same issue is apparent in the SI. Please clarify!

-

Mutant fixation in the presence of a natural enemy

Reply to the referee reports

We would like to thank the two referees and the editor for carefully reading the paper, and coming up with very useful insights and suggestions. We have now implemented all of the suggestions, which have helped with the manuscript's clarity. Below please find a point-by-point reply, where the referees' comments are in blue. To easily see the revisions in the paper, we append versions of the manuscript and supplement with marked changes below this reply.

Reviewer 1

I think this paper is well-written and novel enough and is totally capable of being published in the nature communication journal. However, I have some questions and I would appreciate it if the authors make them clear to me.

Thank you for the positive assessment.

1) In the section 2.2 line, I don't understand what you mean by stochastic Gillespie of the appropriate ODE. Do you use the parameters obtained from ODE in the simulation? Or does it mean the dynamics within the patches deterministic but the migration or the dynamics between the patches is stochastic?

We thank the referee for bringing this up. The stochastic algorithm was not described in the previous version. We have now clarified this by adding a section (Section 1.1) in the new version of the Supplementary Information, where we describe the Gillespie simulation in detail. We point towards this in the main text. The simulations are fully stochastic, using the propensities of different events from their rates in the ODEs. All processes, including local dynamics within a deme, and migration processes, are stochastic. We further specify in more detail the initial conditions, boundary conditions, and the mutant placement algorithm.

OK 2) In Eqs. 25 and 26 of supplementary information I guess what should contribute to the fitness of the mutant should be the success of the mutant and not its failure. Shouldn't it be for example: $w_y^Y P_{x \rightarrow y}$ instead of $w_x^X P_{y \rightarrow x}$?

It is absolutely correct that success of the mutant (in relation to that of the wild type) defines the mutant fitness. This is reflected in equation (29) of the new version of the Appendix (it was unnumbered in the previous version). There, exactly as the referee points out, we have $w_y^Y P_{x \rightarrow y}$ in the expression for the selection coefficient associated with patch conversion.

Equation (27) defines the rate at which conversion takes place, which in turn determines the importance of conversion's contribution to fitness, compared to the process of extinction-colonization, see equation (28) in the updated file. Please note that both expressions (27) and (28) are "out" rates, as they stand for the conversion out of Y and extinction of Y (through infection). In the case where the quantities in equation (29) are small compared to 1, the expressions in (27) and (28) can be replaced by the "in" rates, and the difference will be small in s_p^{conv} and $s_p^{ext-rec}$. (In this case, in (27) we will have $w_y^Y P_{x \rightarrow y}$ instead of $w_x^X P_{y \rightarrow x}$, but as

mentioned before, this is not the expression that defines fitness, but instead, the relative contribution of the conversion process).

We have rewritten the relevant parts of Section 2.3 to clarify these important points.

3) In the Fig. S4 I wonder what is different between mutants and wild-types. Apparently, they both have the same death and reproduction rates.

Thank you for pointing this out. The text has been corrected to reflect the mutant advantage (2%) in division rates.

4) In the Fig. S4 does each frame constitute of 10,000 updating steps? If so, why did you make this setting?

We have rephrased this part of the description, because it was confusing. Now, the time-series in figure S4 are redrawn as functions of time, t (and it is specified that the output began once the mutants reached 20% of the total population). The last panel (panel (d)) contains the statistics of the mutant fractions in all the patches during this simulation. It represents the mutant fractions collected in the course of the simulation, sampled 500 times (after every 10,000 Gillespie updates, which is approximately 1 time unit) during the time-interval depicted in panels (a-c). The text has been updated to include this information.

5) How do you calculate the conversion rate $P_{x \rightarrow y}$ or $P_{y \rightarrow x}$? I didn't see anywhere in the paper mentioning this.

This information is now added at the end of Section 2.2.

And finally, two minor comments:

1) In the supplementary information section 2.2 equations 20-21 parameter Z does not have a subscript in some cases.

This has been corrected, thank you for pointing this out!

2) In the equation after Eq. 26 I think the subscript of s should be p . Am I right?

Yes, this is correct (with "p" standing for "patch" selection coefficient). Thank you for spotting this typo.

Reviewer 2

The manuscript connects two very different fields: The very abstract field of evolutionary graph theory (with the probability of fixation as a central object) and ecological interactions in the form of predator-prey or pathogen host dynamics.

In this kind of ecological setting, a lot of people will think of an internal equilibrium between U and I . The catch here is that there are two types of U , wild type and mutant. It took me some time to grasp this, it is a very nice approach, but I am not sure if this could not be spelled out more clearly. In addition, the model is very different from the usual fixation probability considerations on graphs, as the population size varies – this would correspond to a dynamically changing graph, which is

challenging. Thus, it is very nice that the authors keep the spatial structure relatively simple. I think this is a great piece of work which opens the door for both associated fields to widen their scope. I would have many ideas how to build on this and I would be happy to see this paper published in a prominent place.

Thank you for the positive assessment. We have now explained these concepts more clearly in section 2.1

However, I have a number of issues with this model (actually, they all seem minor to me) that the authors could consider in a revision:

- Would it make sense to show also the associated mean field theory? From simulations, I see fluctuations in the population sizes, but I was wondering if I am correct in translating this into a predator-prey system with a carrying capacity for the prey, which would have a stable internal fixed point? Can you calculate an approximation for Nu from this?

It is true that a corresponding mean field equivalent can in principle be analyzed, limited by a carrying capacity for the prey. In the absence of spatial structure, however, it is known that these systems are characterized by a tendency to show highly oscillatory dynamics. Deterministic models with carrying capacity are characterized by damped oscillations towards a stable equilibrium. Translating this into stochastic simulations, however, leads to the oscillatory dynamics, and stochastic fluctuations around the equilibrium tend to result in population extinctions over wide parameter regions. It was not possible to find a mean field equivalent of the spatial system (with the same number of uninfected cells and infection rate) in which populations persisted in stochastic simulations. It is possible to look at other parameter regions (with much weaker infection / predator activity), in which population persistence is observed. This was addressed in the original manuscript in the following paragraph in the Discussion section, which we have now modified for clarity:

“It is not possible to directly compare the evolutionary dynamics in the spatial systems analyzed here to a corresponding non-spatial setting. For an equivalent parameter set, the dynamics in the absence of spatial population structure result in population extinction. For other parameter regions, where the system persists in the absence of spatial structure, it is possible to investigate how infection influences the fixation probability of the mutant in a non-spatial system. In this case, the infection introduces an increase in demographic fluctuations around equilibrium. Previous work has shown that the existence of demographic fluctuations can independently result in weakened selection (compared to a constant population Moran process)²⁰. However, this is a different mechanism, and based on preliminary simulations, selection is only weakened modestly compared to the magnitude of the effect observed in our spatial dynamics. The reason is that population persistence in the absence of spatial structure is only possible for relatively slow infection rates, which only introduce a limited increase in demographic fluctuations. A more detailed and comprehensive examination of the effect of an infection on demographic fluctuations and mutant fixation in non-spatial systems is subject to future work.”

Can I think about the dynamics between uninfected, infected and empty space as something resembling and SIRS model? Maybe that analogy would work for the infection case and help some readers to grasp what happens here.

In terms of infection models, this would be an “SI” model. Once infected, the individual remains infected for the duration of its life. This resembles an infection of cells by viruses, such as phages

infecting bacteria, which usually do not recover from infection. However, it is also closely related to predator-prey systems, where a predator attack shortens the life of the prey. We have now added a paragraph in the discussion to explain more clearly how our model fits into the generalized enemy-victim dynamics and how it relates and differs from SIS and SIRS models. See the second paragraph of the discussion.

The role of infected cells in the mutation process is not entirely clear to me. If I understand correctly, mutations can only occur in uninfected (=dividing) cells. Can I consider the infected cells as agents that effectively increase death rate of their neighbors, but otherwise not relevant? Is that the reason to discard infected cell from the population size when comparing fixation probabilities? But if there are many infected cells, the population is prone to decline fast – so their presence cannot fully be discarded. Is that another reason the selection appears weaker with infection?

The reason that we have to discard the infected cells from the population size when comparing fixation probabilities is that in the models considered in our study, infected cells do not reproduce. They are essentially dead (it is just a matter of time until they are dead). It is true that an infected individual can effectively increase the death rate of others by infecting them. This however is implicitly taken into account by the fact that we consider the equilibrium population size of uninfected individuals, where all processes are balanced. The situation would be different in an alternative model, where infected cells can reproduce and pass on the virus “vertically” through division. This scenario is beyond the scope of the current study, but would be interesting to investigate in a follow-up study. We have now clarified this in the revised text. First, we clarified why infected cells are not taken into account when calculating fixation probabilities in the agent-based model section 2.1. Further, we discussed alternative assumptions in the second paragraph of the discussion section.

I found section 2.1 under the heading “results” a bit awkward. I would have preferred to have this as its own section on the model as such, but I understand that this is sometimes not possible due to unlucky formatting requirements driven entirely by the needs of experimental biology.

In principle we agree with this. We could move the agent-based model description into its own “Model” section before results. However, we subsequently consider the deme model, and we think it is easier for the reader to follow if the deme model is described just before the deme model results are presented, rather than in a separate section together with the agent-based model. So with this in mind we left it as it was, but would be happy to change if reviewers and editors think otherwise.

In section 2.1, a bit more structure could help, e.g. have bullet points for the different events.

Yes, we agree and have done this in the revised version.

I also found the notation a bit unlucky, maybe the probabilities can have more intuitive names? Is it the idea that the probabilities in the agent based models have capital letters (e.g. “A”) and the associated rates in the ODE small letters (e.g. “a”)? if so, please write that.

Yes, that was the idea. We have now clarified this in the revision (section 2.2).

On p.4, you write about “continuous movement of cells”, but according to the basic model descriptions, cells do not move. Please clarify!

Yes, thank you for pointing this out. We meant to say “spatial separation of uninfected cells from sources of infection, due to cell divisions to adjacent spots, leads to the persistence of the populations on a global level (across the entire grid).” We have corrected this.

When you introduce several mutants, how do you place them? I assume that this makes a major difference.

For the agent-based model, we replace several randomly selected uninfected wild-type cells with mutant cells, at quasi-equilibrium at the same time. This initial mutant placement occurs at a moment in the simulation when the number of uninfected wild-type cells equals the equilibrium value, N_u , determined numerically by calculating the long-term temporal average of the population size. We have now expressed this more clearly in the revision, see section 2.1 of the main text.

For the deme model, the following procedure was used: We introduced the mutants when the total global population size of uninfected cells at quasi-equilibrium was equal to the rounded temporal average of the global population, N_u . To introduce a mutant, a random patch was selected with the probability proportional to its number of uninfected wild-type individuals, and in that patch, a single uninfected wild-type individual was replaced by a mutant one. To introduce multiple mutants, this procedure was repeated as many times as the desired number of mutants. We have explained this now in the main text, section 2.2. Further details about the deme model simulations, including initial conditions and boundary conditions, are provided in the revised supplementary materials, section 1.1.

“Significantly reduced” – I am sure you can make it significant by simulating more, but I suggest to drop this word as some people think p-values when they see significant.

We agree and replaced “significantly” with strongly, substantially, noticeably”.

“stochastic Gillespie simulations of the appropriate ODEs“ is very complex. I assume that you take the rates of an individual based process and use them either to derive a deterministic ODE or to do Gillespie-type simulations. In principle, there are many individual based rate processes that lead to a single ODE, so I do not think one can just say that we work on the corresponding stochastic model. The same issue is apparent in the SI. Please clarify!

We thank the referee for bringing up this important issue. In the previous version of the manuscript, the stochastic algorithm was not clearly defined. In the new Section 1.1 of the Supplementary Information we describe in detail the version of a Gillespie algorithm that we use in our simulations. The Gillespie simulations of ODEs are used in the context of the deme model, where the ODE’s terms are used as propensities of the various events in the stochastic system. In the supplement, we further specify in more detail the initial conditions, boundary conditions, and the mutant placement algorithm. In the main text, we point to the supplement for further explanation.

Mutant fixation in the presence of a natural enemy

Dominik Wodarz^{1,2} and Natalia L. Komarova²

1: Department of Population Health and Disease Prevention, University of California, Irvine, CA 92697

2: Department of Mathematics, University of California, Irvine, CA 92697

Abstract

In the extensive literature about mutant invasion and fixation, populations are typically assumed to exist in isolation from their ecosystem. Yet, populations are part of ecological communities, and enemy-victim (e.g. predator-prey or pathogen-host) dynamics are particularly common. We use computational models to re-visit the established theory about mutant fixation in the presence of a natural enemy, which equally attacks both wild-type and mutant populations. We consider advantageous and disadvantageous mutants, whose fitness is unrelated to the infection. Using spatially structured agent-based as well as patch models of a population that is subject to infection, we investigate the fixation probability of a mutant that is introduced into the population at quasi-equilibrium. We find that infection substantially weakens selection. Thus, the presence of infection increases the fixation probability of disadvantageous mutants and decreases the fixation probability of advantageous mutants, with the magnitude of the effect rising with the infection rate. We show that this occurs because infection induces spatial structures, in which mutant and wild-type individuals are mostly spatially separated. Thus, instead of mutant and wild-type individuals competing with each other, it is mutant and wild-type “patches” that compete, resulting in smaller fitness differences and hence weakened selection. Because natural enemies such as infections are ubiquitous, this has broad applicability to natural systems. Our results imply that the burden of deleterious mutants in natural populations might be substantially higher than expected from mutant invasion theory developed in the absence of natural enemies.

Deleted: significantly

Deleted: strongly weakens

Deleted: significantly

1. Introduction

In reproducing populations, evolution is driven by the generation of new mutations, and the fate of the mutants is determined by selection and drift. The dynamics of mutant invasion have been studied extensively in a variety of settings^{1,2}, driven in large part by the analysis of mathematical models. The fixation probability of a mutant is a central concept in this respect^{1,3,4}. It is defined as the probability for a mutant that has been introduced into a population to rise and replace the wild-type. The conditional average time to fixation of a mutant is another important measure, determined across those instances where mutant fixation occurs. An extensive literature exists assuming constant populations^{3,5-7}, which can be described mathematically by e.g. the Moran process or the Fisher-Wright process. Much of this work assumes well-mixed populations, although important insights into the dynamics of mutant invasion have been obtained for spatially or deme-structured populations⁸⁻¹³, as well as more generally for mutant fixation on graphs¹⁴⁻¹⁸. Besides constant populations, the effect of demographic fluctuations around an equilibrium on the probability of mutant fixation has also been analyzed¹⁹⁻²⁴.

Evolutionary theory about mutant fixation has typically focused on the evolving population in isolation, which has given rise to many fundamental insights. In nature, however, evolving populations exist as part of an ecosystem. Natural enemies present a particularly common ecological setting. Yet, it is currently unclear how the presence of a natural enemy (that equally attacks both wild-type and mutant individuals) influences the fixation probability of a mutant. Within such a system, the evolving population can persist around an equilibrium, which at first sight seems similar to a constant population scenario. In spatially structured (and hence biologically realistic) models, however, the stable persistence of the population can be the result of continuous local extinction events coupled with migration of individuals into temporary refuge spaces without enemies, as illustrated by patch and metapopulation models²⁵. Population fluctuation, frequent extinction events, and bottlenecks have been shown to change the properties of mutant invasion^{20,26,27}, and hence it is important to study the spatial dynamics of mutant invasion in the presence of a natural enemy.

Here, we study the properties of mutant invasion and fixation in spatially structured populations at quasi-equilibrium, assuming that the evolving population exists in the presence of a natural enemy. While applicable to all enemy-victim settings, the model is formulated as a population of cells that are subject to infection by a virus (regardless of cell genotype). We start by considering a spatial stochastic agent-based model and then compare its properties to those of patch models.

2. Results

2.1. Spatial agent-based model of host evolution in the presence of infection

We consider an agent-based model (ABM) on a 2D grid, where each of $n_1 \times n_2$ spots could be either empty or contain an uninfected or infected cell of different types (wild-type or mutant). Each time-step consists of N_t elementary updates, where N_t is the number of currently occupied sites. At each elementary update, a random cell is picked, and the following events can occur:

- If an uninfected cell is picked it can attempt division with a probability R . A random spot among the 8 nearest neighbors is chosen, and if unoccupied, the offspring cell is placed there.
- With a probability D , the uninfected cell dies.
- With probability $1-R-D$, no event occurs for the uninfected cell.
- Infected cells are assumed not to divide. When selected, they can die with a probability A .
- Infected cells attempt an infection event with a probability B , during which a random spot is chosen among the 8 nearest neighbors. The infection proceeds if the chosen spot contains an uninfected cell.
- With probability $1-A-B$, no event occurs for infected cells.

Periodic boundary conditions were used in all simulations.

In the absence of mutants, i.e. just one cell type in a spatial pathogen-host system, the dynamics have been well defined. Over time, the population sizes of

Deleted: d.

Deleted: For uninfected cells, the following events can occur.

Deleted: W

Deleted: L

Deleted: the

Deleted: cell attempts a division.

Formatted: Font: (Default) Arial

Formatted: Font: (Default) Arial

Formatted: Font: (Default) Arial

Formatted: Font: (Default) Arial

Formatted: Font: (Default) Arial

Formatted: Font: (Default) Arial

Deleted: ,

Formatted: Font: (Default) Arial

Deleted: and w

Formatted: Font: (Default) Arial

Deleted: L

Formatted: Font: (Default) Arial

Formatted: Font: (Default) Arial

Formatted: Font: (Default) Arial

Deleted: ,

Formatted: Font: (Default) Arial

Deleted: and

Formatted: Font: (Default) Arial

Formatted: Font: (Default) Arial

Formatted: List Paragraph, Bulleted + Level: 1 + Aligned at: 0.25" + Indent at: 0.5"

Formatted: Font: (Default) Arial

uninfected and infected cells converge to a state where they fluctuate around an equilibrium (Fig 1); we will refer to the mean equilibrium size of the uninfected population as N_u . The spatial distribution of cells, however, strongly depends on the rate of infection. For relatively low infection rates, the cells are distributed more uniformly through space (Fig 1A). For higher infection rates, however, pronounced spatial structures emerge in which moving fronts of uninfected cells are “chased” by infected cells (Fig 1B). In a particular local area, the infection drives the cell population extinct.

Spatial separation of uninfected cells from sources of infection, due to cell divisions to adjacent spots, leads to the persistence of the populations on a global level (across the entire grid). This recapitulates the well-known spatial refuge effect that can contribute to population persistence and spatial pattern formation in predator-prey dynamics ²⁵.

Formatted: Font: 12 pt

Deleted: Continuous movement of cells, through division into adjacent spots, however, leads to the persistence of the populations on a global level (across the entire grid)

Figure 1: Basic dynamics of infected and uninfected cells without evolution. For the time series, light blue and yellow colors represent the populations of uninfected and infected cells. For the spatial pictures, light blue and yellow colors also represent uninfected and infected cells. Dark blue shows empty space. (A) Low infection rate. (B) High infection rate. $R=0.5$; $D=0.05$; $A=0.1$; $n1=n2=100$. For panel (A) $B=0.2$; for panel (B) $B=0.9$.

Deleted: L

To study the fixation probability and conditional fixation times of mutant cells, we start with a square block of uninfected wild-type individuals in the middle of the grid and place a smaller square block of infected wild-type individuals inside this block. We then simulate population dynamics in the absence of mutants for a certain amount of time, until quasi-equilibrium is reached (the exact choice of the initial condition is unimportant for reaching this state). Next, we replace one (or several) randomly selected uninfected wild-type cells with a corresponding number of mutant cells, at quasi-equilibrium. This initial mutant placement occurs at a moment in the simulation when the number of uninfected wild-type cells equals the equilibrium value, N_u , determined numerically by calculating the long-term temporal average of the population size. If several instead of one mutant cell are introduced, multiple randomly chosen wild-type uninfected cells are turned into mutants at the same time. No *de novo* mutations are considered. We then allow the simulation to run until either the mutant is extinct, or until the mutant population has replaced the wild-type cell population (mutant fixation). The simulation is run repeatedly and the fraction of runs during which mutant fixation occurs is determined. For the cases of mutant fixation, the time until fixation is determined (conditional fixation time).

This setup unites evolutionary theory on graphs¹⁴⁻¹⁸ with pathogen-host or predator-prey dynamics. As a consequence, we are not studying mutant evolution in a constant population, but in a population that changes dynamically (both temporally and spatially), driven by the pathogen-host interactions. Thus, before mutant introduction, uninfected and infected populations fluctuate around a steady state, with the fluctuations being more pronounced for higher infection rates. Mutant uninfected cells are introduced into this system, and the evolutionary fate of the mutant uninfected cells is followed. Mutant infected cells do not contribute to mutant spread in this model, because infected cells are assumed not to divide, and to die after a certain period of time.

These dynamics are investigated for neutral, advantageous, and disadvantageous mutants, comparing the results in the absence and presence of

Deleted: first

Deleted: u

Deleted: W

Deleted: then

Deleted: fast

infection, where the rate of infection is varied. To implement fitness advantage/disadvantage of mutants, we assume that the division rate of cells is increased/decreased relative to the wild-type by multiplying it with the coefficient $(1+s)$, where s is the selection coefficient. An advantageous mutant corresponds to $s>0$, and for a disadvantageous mutant $s<0$.

When comparing the dynamics under different infection rates, equilibrium population sizes vary. To control for this, the grid size is adjusted such that the average equilibrium population size of uninfected cells, N_u , is kept approximately constant, regardless of infection rate. Probabilities of mutant fixation in the presence of infection are compared with those in the absence of infection, and also with the well-known formula for the Moran process of equivalent size, $P_{\text{fix}}(i) = \frac{1-1/(1+s)^i}{1-1/(1+s)^n}$, where i denotes the initial number of mutants in a total population of n individuals (N_u in our setting).

For neutral mutants, as expected, the numerically obtained fixation probability is $1/N_u$ regardless of infection rate. This is identical to the fixation probability provided by the constant population Moran process, with the population size given by the number of uninfected cells. Only the population of uninfected cells determines the fixation probability because the infected cells are assumed to not divide in our model. Therefore, once infected, the cells are destined to die in time, determined by the death rate of infected cells, A .

For advantageous mutants, we find that the presence of infection weakens selection (Fig 2A). Without infection, the numerically obtained mutant fixation probability is very close to the value predicted by the non-spatial Moran process (for a discussion of the role of demographic fluctuations see ref ²⁰); therefore, here and in other cases below, the Moran formula is a convenient reference point for evaluating the changes in fixation probability due to infection. In the presence of the virus, the fixation probability is noticeably reduced, with larger reductions observed for faster infection rates. The

Deleted:

Deleted: given

Deleted: significantly

Deleted: noticeably

average conditional time to fixation is found to be lower in the presence compared to the absence of infection, with shorter fixation times occurring for faster infection rates.

For disadvantageous mutants, selection is again found to be weakened in the presence of the virus (Fig 2B). Without infection, the numerically obtained mutant fixation probability is again close to the value predicted by the non-spatial Moran process. In the presence of infection, however, the fixation probability of disadvantageous mutants is noticeably increased, with the larger increases seen for faster infection rates. An up to 1000-fold increase in the fixation probability is seen for the parameter regime studied here (Fig 2B). The average conditional time to fixation is again shortened by the virus (Fig 2B).

Deleted: significantly

Deleted: noticeably

Figure 2. Fixation probabilities (with 95% confidence limits) and conditional fixation times (with standard errors) in the agent-based model for different infection probabilities. (A) Advantageous mutants for two different values of selection coefficient, s . 1 mutant was introduced. (B) Disadvantageous mutants for two different values of selection coefficient, s . 500 mutants were introduced. Parameters are as follows. $R=0.5$; $D=0.05$; $A=0.1$; $N_i=944$. The grid sizes for the successive infection probabilities are: 32x33, 42x42, 60x60, 73x73, 81x80.

2.2. Deme models of host mutant evolution in the presence of infection

An alternative and coarser grained method of modeling spatial interactions are deme or patch models. Rather than tracking each individual and their spatial location, populations in the demes are assumed to be well-mixed. In addition, individuals migrate between patches, and migration can be spatially restricted to nearest neighbors, or less spatially restricted. Here we consider a two-dimensional deme / patch model, consisting of $n_1 \times n_2 \equiv \mathcal{N}$ patches. In each patch, host-pathogen dynamics are described by stochastic Gillespie simulations of ODEs, given by

$$\frac{dx_i}{dt} = rx_i \left(1 - \frac{x_i + z_i}{K}\right) - \beta x_i z_i + \mu \left(\frac{\sum_k x_k}{\mathcal{N}} - x_i\right),$$

Deleted: L

Deleted: ,

Deleted: and their dynamics within a deme are described by stochastic Gillespie simulations of the appropriate ODEs.

Formatted: Font: (Default) Arial

$$\frac{dz_i}{dt} = \beta x_i z_i - a z_i + \mu \left(\frac{\sum_k z_k}{N} - z_i \right),$$

where x_j and z_j denote the populations of susceptible and infected cells in patch j . Lower case letters are used for rate parameters, as opposed to the capital letters used for probabilities in the agent-based model. The parameter r is the basic division rate of cells, d is the death rate of susceptible cells, β is the rate of infection, and a stands for the death rate of infected cells. Migration of uninfected and infected cells to/from other patches occurs with a rate μ , and migration can occur either to/from any patch in the system (non-spatial migration, shown in the equations above), or to/from the eight nearest neighbors (spatial migration, see Supplementary Materials Section 1.1). K is the carrying capacity of a patch. For stochastic simulations of mutant spread, the Gillespie algorithm is applied to this patch ODE model with details given in the Supplementary Materials (Section 1.1). When varying the infection rate of the virus, the grid size is again adjusted to maintain approximately the same total number of uninfected cells across the different infection rates.

Over time, the sum of populations in this model fluctuates around a steady state value (Fig. 3). Although there is a global quasi-steady state, within each patch, populations can crash to extinction due to the infection, and can subsequently be re-colonized to repeat this pattern²⁵. To investigate mutant invasion, we introduced the mutants when the total global population size of uninfected cells is equal to the rounded temporal average of the global population, N_u . To introduce a mutant, a random patch was selected with the probability proportional to its number of uninfected wild-type individuals, and in that patch, a single uninfected wild-type individual was replaced by a mutant one. To introduce multiple mutants, this procedure was repeated as many times as the desired number of mutants. The fixation probabilities and times were determined in the same way as for the ABM.

Deleted: $dS/dt = rS(1-(S+I)/K) - \delta S - \beta SI$; $dI/dt = \beta SI - aI$,

Deleted: [add migration terms in generality – spatial or non-spatial]

Formatted: Highlight

Deleted: S

Deleted:

Deleted: n

Formatted: Highlight

Deleted: /

Formatted: Font: Italic

Deleted: .

Deleted: T

Deleted: K is the carrying capacity of a patch, δ

Deleted: m

Formatted: Font: Italic

Deleted: or to/from any patch in the system (non-spatial migration)

Formatted: Not Highlight

Deleted: s

Deleted: In addition to these processes, the Gillespie simulations assume that uninfected and infected cells can migrate to a randomly chosen destination patch with a rate m . With spatial migration, the destination patch is chosen randomly from the eight nearest neighboring patches. For non-spatial migration, the destination patch is chosen randomly from any of the patches in the system.

Deleted:

Deleted: ¶

Figure 3. Dynamics in the patch model with migration to nearest neighboring patches. Blue and orange colors represent the populations of uninfected and infected cells. (A) Dynamics within a patch. (B) Total dynamics, with population sizes summed up across all patches. Parameters are as follows: $r=0.7$; $d=0.1$; $\beta=0.5$; $a=0.5$; $K=100$, $\mu_s=0.02$, $n_1=n_2=19$.

Deleted: δ

Deleted: m

We observe results that are qualitatively similar to the ABM described above. That is, selection is weakened both for advantageous and disadvantageous mutants (Fig 4): the fixation probability of advantageous mutants decreases below the value predicted by the non-spatial Moran process as the infection rate rises, and the fixation probability of disadvantageous mutants increases above that of the Moran process. The conditional time to fixation is generally again reduced by the presence of the infection (Fig 4), although for spatial migration the dependence can be non-monotonous. In previous work, conditional fixation times have been shown to follow complex patterns²⁸, and it is beyond the scope of the current study to investigate this in further detail. It is interesting to note that these trends apply both to simulations with spatial and non-spatial migration (Fig 4).

Deleted: ¶

Formatted: Indent: First line: 0"

Figure 4. Fixation probabilities (with 95% confidence limits) and conditional fixation times (with standard errors) in the deme model with spatial and non-spatial migration. (A) Advantageous mutants, $s = 0.02$; 20 mutants were introduced. (B) Disadvantageous mutants, $s = -0.001$; 9000 mutants were introduced. Parameters were as follows: $r=0.7$; $c=0.1$; $a=0.5$; $k=100$, $\mu=0.02$; $Nu=11288$. For spatial migration, the grid sizes for the successive infection rates are 11x12, 14x14, 22x21, 22x22, 20x21, 20x19. For non-spatial migration, they are 11x12, 14x14, 22x22, 24x24, 23x24, 22x23.

Deleted: δ
Deleted: m

2.3. Patch versus cell competition

We propose that the reason for the weakened selection observed in the presence of infection is the behavior of the system as a metapopulation, regardless of the underlying model. That is, cells go extinct locally as a result of the infection, and persist by colonizing other areas of space, which temporarily do not contain infection and hence provide a refuge for the cells. This happens across a continuous space in the ABM, as shown in Fig 1. It happens more explicitly in the patch models where patches periodically go extinct and become recolonized (Fig 3), both under spatial and non-spatial migration. For relatively large infection rates, this also leads to a spatial

separation of wild-type and mutant cells. In terms of the patch model, a patch is likely to either contain only wild-type cells or only mutant cells, but rarely both. In this setting, mutant and wild-type patches (rather than cells) effectively compete for colonization of empty patches, and this leads to mutant fixation probabilities that deviate noticeably from those predicted by the Moran model (or the process without infection). For low infection rates or in the absence of infection, mixing of wild-type and mutant cells is more likely, and it is the competition among cells (rather than patches) that drives mutant fixation. Consequently, the observed fixation probabilities converge to those predicted by the Moran process. For intermediate infection rates, the fixation probabilities are determined by a mixture of cell and patch competition.

Deleted: significantly
Deleted: noticeably

Figure 5. A schematic explaining the components of mutant dynamics in patches. (a) Populations of patches can be uninfected wild-type (X), uninfected mutant (Y), and infected (Z). (b) Coarse-grained model structure includes processes of colonization, infection, extinction, and conversion. Green and purple dots denote uninfected wild-type and mutant cells, and red dots indicate infected cells.

We demonstrate this in more detail using the patch model with non-spatial migration, see Fig.5. Assuming that mutant and wild-type cells do not co-occur in the same patches (panel (a)), we can write down a coarse-grained model where patches, rather than individual cells, are agents, and where population dynamics are governed by the following processes (panel (b)): empty patch colonization, patch infection, infected patch

extinction, and patch conversion (a cell of the other type migrating into an uninfected patch and taking over). Let us denote by X, Y , and Z the total numbers of uninfected wild-type patches, uninfected mutant patches, and infected patches, and by w_x, w_y , and w_z the mean per-deme populations of these types of cells, in patches of types X, Y , and Z respectively. We can summarize the coarse-grained dynamics as follows:

$$\dot{X} = \mu X w_x \left(1 - \frac{X + Y + Z}{N}\right) P_{col}^x - \mu Z w_z \frac{X}{N} P_{inf}^x - \mu Y w_y \frac{X}{N} P_{x \rightarrow y} + \mu X w_x \frac{Y}{N} P_{y \rightarrow x},$$

$$\dot{Y} = \mu Y w_y \left(1 - \frac{X + Y + Z}{N}\right) P_{col}^y - \mu Z w_z \frac{Y}{N} P_{inf}^y + \mu Y w_y \frac{X}{N} P_{x \rightarrow y} - \mu X w_x \frac{Y}{N} P_{y \rightarrow x},$$

$$\dot{Z} = \mu Z w_z \frac{X P_{inf}^x + Y P_{inf}^y}{N} - \frac{Z}{T_{ext}}$$

Here, the first term in the right hand side of the equations for X and Y describes colonization of empty patches, the second term describes the process of infection (the same terms reappear in the equation for Z with the opposite signs). The remaining terms in the X and Y equations describe conversion, and the last term in the equation for Z is patch extinction (see Supplement for complete details of this model). Denote the equilibrium number of uninfected patches, in the absence of mutants, by X_{eq} . At the level of patch competition, there are two selection coefficients: one associated with patch conversion (s_p^{conv}) and the other with the process of extinction-recolonization ($s_p^{ext-rec}$). The overall selection coefficient is given by the mean of these two, weighted with their respective rates, R_{conv} and $R_{ext-rec}$:

$s_p = \frac{R_{ext-rec} s_p^{ext-rec} + R_{conv} s_p^{conv}}{R_{ext-rec} + R_{conv}}$, and the probability of mutant demes taking over can be

calculated by the Moran formula, applied at the level of patches: $\rho_i =$

$\frac{1 - 1/(1+s_p)^i}{1 - 1/(1+s_p)^{X_{eq}}}$, where i is the initial number of mutant patches. If the majority of demes are

occupied and conversion is the main driving force of mutant dynamics, then the selection coefficient is approximately s_p^{conv} , and the probability of mutant take-over is approximately that predicted by the usual Moran process in the absence of infection (and with the equivalent population size). This is because the process of deme

conversion through death-birth events is the same as what comprises the conventional dynamics that leads to the Moran formula. If, however, the system of patches is sparsely populated and the process of recolonization is the leading force of mutant spread, then we have $s_p \approx s_p^{ext-rec}$, where

$$s_p^{ext-rec} = \frac{w_y P_{col}^y P_{inf}^x}{w_x P_{col}^x P_{inf}^y} - 1, \text{ containing quantities } P_{col}^x = 1 - d_x/r_x \text{ and } P_{col}^y = 1 - d_y/r_y,$$

which are the probabilities for a single wild-type (mutant) cell to successfully colonize a patch, and $P_{inf}^x = 1 - \left(\frac{w_x \beta}{a}\right)^{-1}$ and $P_{inf}^y = 1 - \left(\frac{w_y \beta}{a}\right)^{-1}$, which are the probabilities for a single infected cell to start successful infection in a wild-type (mutant) patch. The selection coefficient corresponding to the process of recolonization has a **much** smaller absolute value compared to s_p^{conv} , and describes much weaker selection. In the Supplement we provide a comparison **of** the prediction of the coarse-grained model with numerical simulations of the full system, and find them in close agreement (in the region of model applicability). While the coarse-grained approach is somewhat limited in that it does not describe patch models with spatial migration, or spatially distributed ABM systems, it provides a way to separate the different processes that contribute to mutant dynamics, and to explain the observation of a significant selection reduction in systems with infection.

Deleted: significantly

Deleted: with

3. Discussion

We have shown that the dynamics of mutant invasion can be **strongly** altered if both wild-type and mutant individuals are equally susceptible to a natural enemy such as an infection. In particular, we showed that this can lead to a significant weakening of selection. Thus, the fixation probability of a disadvantageous mutant can be strongly increased compared to the evolutionary dynamics in the absence of infection, and the fixation probability of advantageous mutants can be **substantially** lowered. The difference can be several orders of magnitude. In the presence of infection, it is possible that a deleterious mutant, whose invasion potential can be neglected according to traditional theory, has a reasonable chance of emerging. These results are highly

Deleted: significantly

Deleted: significantly

relevant for natural settings, because most populations do not exist in isolation, but are part of a larger ecosystem that includes natural enemies.

In this analysis, we considered the simplest infection system, which corresponds most closely to other natural enemy interactions, such as predator-prey or parasitoid-host interactions. Once infected, an individual is destined to die. The infected individual is assumed not to reproduce anymore, or to recover from the infection. Hence, this would represent an SI (susceptible-infected) system. In future work, it would be interesting to extend our framework to more complex infection systems, such as SIR (susceptible-infected-recovered), or SIRS models, or to models in which infected individuals are assumed to reproduce, thereby transmitting the pathogen vertically.

We only analyzed spatially explicit models in this study, because spatial dynamics are well-known to lead to more realistic and stable enemy-victim dynamics, and lack of any spatial population structure typically results in highly oscillatory dynamics that are likely to lead to population extinction in stochastic models. Using a combination of numerical and analytic methods, we have shown that the mechanism underlying the weakened selection in this setting is a result of the population structure assumed in our models. In our system, an increase in the infection rate results in dynamics that are progressively less stable on a local level. That is, local population extinction occurs frequently, and persistence across space is possible because host populations keep dividing or migrating (in the deme model), into new locations where they can grow temporarily without the infection present ²⁵. In this setting, it is not the competition between individuals that matters because a given location rarely contains both mutant and wild-type individuals together. Instead, most locations contain either wild-type or mutants, and patches compete with each other in the context of extinction and recolonization of local areas / patches / demes. This leads to a much lower difference in fitness of mutant compared to wild-type individuals, accounting for the large differences seen in the fixation probabilities. These dynamics are related to extinction-recolonization dynamics that have been explored in the absence of natural enemies, also demonstrating weakened selection ²⁶.

Deleted: moving

It is not possible to directly compare the evolutionary dynamics in the spatial systems analyzed here to a corresponding non-spatial setting. For an equivalent parameter set, the dynamics in the absence of spatial population structure result in population extinction. For other parameter regions, where the system persists in the absence of spatial structure, it is possible to investigate how infection influences the fixation probability of the mutant in a non-spatial system. In this case, the infection introduces an increase in demographic fluctuations around equilibrium. Previous work has shown that the existence of demographic fluctuations can independently result in weakened selection (compared to a constant population Moran process)²⁰. However, this is a different mechanism, and based on preliminary simulations, selection is only weakened modestly compared to the magnitude of the effect observed in our spatial dynamics. The reason is that population persistence in the absence of spatial structure is only possible for relatively slow infection rates, which only introduce a limited increase in demographic fluctuations. A more detailed and comprehensive examination of the effect of an infection on demographic fluctuations and mutant fixation in non-spatial systems is subject to future work.

Deleted: in the presence of infection

Experimental data that could be used to address the theoretical predictions presented here are so far not available. Bacteria-phage interactions²⁹ are a biological system in which these dynamics could be explored. In other contexts, bacteria and phages have been used to study aspects of virus-host evolution experimentally. Phage infections have been shown to have a direct impact on the evolution of the bacterial cell populations³⁰⁻³³. For example, the presence of bacteriophages can increase the evolutionary potential of the bacterial population, which can provide a benefit to the bacteria that balances the increased cell mortality induced by the infection³⁰. In the context of antibiotic resistance evolution in bacteria, phage infections have been shown to result in the emergence of bacterial strains that are resistant to the phage, and interestingly, phage-resistance could lead to the simultaneous increase in bacterial sensitivity to the antibiotic^{34,35}. Other examples of the eco-evolutionary dynamics of phages and bacteria are discussed in references^{29,36,37}.

Our present investigation has focused on very basic evolutionary dynamics, i.e. revisiting the theory on mutant invasion if the population is subject to infection, rather than evolving in isolation. The evolutionary processes considered in our models are not in response to the selection pressure exerted by the infection, but represent infection-unrelated evolutionary changes that can be disadvantageous, advantageous, or neutral. This is in contrast to an extensive literature about host-pathogen co-evolution³⁸ that typically investigates scenarios where the infection is a driver of (co-)evolutionary change. The work presented here, however, has shown that the presence of an infection, or a natural enemy in general, can fundamentally alter the basic dynamics of mutant invasion in a population, through spatial heterogeneity generated by underlying dynamics.

Data availability statement: The results reported in this manuscript are based on mathematical analysis and corresponding computer simulations (and not empirical data). The mathematical analysis is fully described in the main text and the Supplementary Information.

Code availability statement: Computer codes for the agent-based model and the patch models are provided as Supplementary Information.

Formatted: Font: Bold

References

1. Kimura M. On the probability of fixation of mutant genes in a population. *Genetics*. 1962;47:713-719.
2. Kimura M. *Population Genetics, Molecular Evolution, and Neutral Theory: Selected Papers*. Chicago: University of Chicago Press; 1994.
3. Patwa Z, Wahl LM. The fixation probability of beneficial mutations. *Journal of the Royal Society, Interface*. 2008;5(28):1279-1289.
4. Loewe L, Hill WG. The population genetics of mutations: good, bad and indifferent. In: The Royal Society; 2010.
5. Moran PAP. Random processes in genetics. Paper presented at: Mathematical proceedings of the cambridge philosophical society 1958.
6. Moran PAP. *The statistical processes of evolutionary theory*. Clarendon Press; 1964.
7. Shafiey H, Waxman D. Exact results for the probability and stochastic dynamics of fixation in the wright-fisher model. *Journal of theoretical biology*. 2017;430:64-77.
8. Whitlock MC. Fixation probability and time in subdivided populations. *Genetics*. 2003;164(2):767-779.
9. Hauert C, Imhof LA. Evolutionary games in deme structured, finite populations. *Journal of theoretical biology*. 2012;299:106-112.
10. Parvinen K. Evolution of migration in a metapopulation. *Bulletin of mathematical biology*. 1999;61(3):531-550.
11. Wakeley J, Takahashi T. The many-demes limit for selection and drift in a subdivided population. *Theoretical population biology*. 2004;66(2):83-91.
12. Hanski I. Metapopulation dynamics. *Nature*. 1998;396(6706):41-49.
13. Yagoobi S, Traulsen A. Fixation probabilities in network structured meta-populations. *Scientific Reports*. 2021;11(1):17979.
14. Lieberman E, Hauert C, Nowak MA. Evolutionary dynamics on graphs. *Nature*. 2005;433(7023):312-316.
15. Möller M, Hindersin L, Traulsen A. Exploring and mapping the universe of evolutionary graphs identifies structural properties affecting fixation probability and time. *Communications biology*. 2019;2(1):137.
16. Hindersin L, Möller M, Traulsen A, Bauer B. Exact numerical calculation of fixation probability and time on graphs. *Biosystems*. 2016;150:87-91.
17. Allen B, Sample C, Steinhagen P, et al. Fixation probabilities in graph-structured populations under weak selection. *PLoS computational biology*. 2021;17(2):e1008695.
18. Antal T, Redner S, Sood V. Evolutionary dynamics on degree-heterogeneous graphs. *Physical review letters*. 2006;96(18):188104.
19. Campbell R. A logistic branching process for population genetics. *Journal of theoretical biology*. 2003;225(2):195-203.
20. Parsons TL, Quince C. Fixation in haploid populations exhibiting density dependence I: the non-neutral case. *Theoretical population biology*. 2007;72(1):121-135.
21. Engen S, Lande R, SAETHer B-E. Fixation probability of beneficial mutations in a fluctuating population. *Genetics research*. 2009;91(1):73-82.

22. Otto SP, Whitlock MC. The probability of fixation in populations of changing size. *Genetics*. 1997;146(2):723-733.
23. Pollak E. Fixation probabilities when the population size undergoes cyclic fluctuations. *Theoretical population biology*. 2000;57(1):51-58.
24. Czuppon P, Traulsen A. Fixation probabilities in populations under demographic fluctuations. *J Math Biol*. 2018;77(4):1233-1277.
25. Hassell MP, Comins HN, Mayt RM. Spatial structure and chaos in insect population dynamics. *Nature*. 1991;353(6341):255-258.
26. Cherry JL. Selection in a subdivided population with local extinction and recolonization. *Genetics*. 2003;164(2):789-795.
27. Bittihn P, Hasty J, Tsimring LS. Suppression of beneficial mutations in dynamic microbial populations. *Physical review letters*. 2017;118(2):028102.
28. Hindersin L, Traulsen A. Counterintuitive properties of the fixation time in network-structured populations. *Journal of The Royal Society Interface*. 2014;11(99):20140606.
29. Koskella B, Brockhurst MA. Bacteria–phage coevolution as a driver of ecological and evolutionary processes in microbial communities. *FEMS microbiology reviews*. 2014;38(5):916-931.
30. Williams HT. Phage-induced diversification improves host evolvability. *BMC evolutionary biology*. 2013;13(1):1-17.
31. Tazyman SJ, Hall AR. Lytic phages obscure the cost of antibiotic resistance in *Escherichia coli*. *The ISME journal*. 2015;9(4):809-820.
32. Harcombe W, Bull J. Impact of phages on two-species bacterial communities. *Applied and Environmental Microbiology*. 2005;71(9):5254-5259.
33. Joo J, Gunny M, Cases M, Hudson P, Albert R, Harvill E. Bacteriophage-mediated competition in *Bordetella* bacteria. *Proceedings of the Royal Society B: Biological Sciences*. 2006;273(1595):1843-1848.
34. Burmeister AR, Fortier A, Roush C, et al. Pleiotropy complicates a trade-off between phage resistance and antibiotic resistance. *Proceedings of the National Academy of Sciences*. 2020;117(21):11207-11216.
35. Chan BK, Sistro M, Wertz JE, Kortright KE, Narayan D, Turner PE. Phage selection restores antibiotic sensitivity in MDR *Pseudomonas aeruginosa*. *Scientific reports*. 2016;6(1):1-8.
36. Blazanin M, Turner PE. Community context matters for bacteria-phage ecology and evolution. *The ISME Journal*. 2021;15(11):3119-3128.
37. Bohannan BJ, Lenski RE. Linking genetic change to community evolution: insights from studies of bacteria and bacteriophage. *Ecology letters*. 2000;3(4):362-377.
38. Buckingham LJ, Ashby B. Coevolutionary theory of hosts and parasites. *Journal of Evolutionary Biology*. 2022;35(2):205-224.

Mutant fixation in the presence of a natural enemy

Supplementary information

Contents

1	A coarse-grained approach in the absence of mutants	1
1.1	The patch model of infected and uninfected cells and Gillespie simulations	1
1.2	A coarse-grained description	3
1.3	Stochastic simulations and coarse-grained predictions	7
2	A coarse-grained approach in the presence of mutants	9
2.1	The patch model in the presence of two types of hosts	9
2.2	A coarse-grained system in the presence of two types of hosts	9
2.3	Quantification of selection dynamics	11
2.4	Comparison with simulations	14

1 A coarse-grained approach in the absence of mutants

1.1 The patch model of infected and uninfected cells and Gillespie simulations

Consider the patch model described in the main text. We first focus on the basic dynamics in the absence of mutants. There are \mathcal{N} patches, and cells migrate between patches randomly and uniformly, regardless of their location. Denote by x_i and z_i the populations of uninfected and infected cells in deme i , respectively. The co-dynamics of cells and infection within patches happen according the following ODEs:

$$\dot{x}_i = rx_i \left(1 - \frac{x_i + z_i}{K}\right) - dx_i - \beta x_i z_i + \mu \left(\frac{\sum_{k=1}^{\mathcal{N}} x_k}{\mathcal{N}} - x_i\right), \quad (1)$$

$$\dot{z}_i = \beta x_i z_i - az_i + \mu \left(\frac{\sum_{k=1}^{\mathcal{N}} z_k}{\mathcal{N}} - z_i\right), \quad 1 \leq i \leq \mathcal{N}, \quad (2)$$

where r and d are respectively the linear growth and death rates of uninfected cells, K is the demes' carrying capacity, β is the infection rate, a is the death rate of infected cells, and μ is the migration rate of cells. Equations (1-2) describe the version of the model characterized by non-spatial migration, that is, migration happens among all the \mathcal{N} patches. To study the nearest-neighbor migration, we would replace the term containing μ in equation (1) with

$$\sum_{k=1}^{\mathcal{N}} \mu_{ik}(x_k - x_i),$$

and similarly, for equation (2), where $\{\mu_{ik}\}$ is a symmetric migration matrix. To implement migration to the nearest neighbors, we placed the patches on a square grid with periodic boundary conditions, and set $\mu_{ij} = \mu/8$ if patches i and j are neighbors in the Moore neighborhood sense, and $\mu_{ij} = 0$ otherwise.

Here we will illustrate how the stochastic Gillespie simulations were implemented using the example of system (1-2). For convenience, in the stochastic system, let us suppose that the current number of uninfected individuals in patch i is denoted by x_i , and the current number of infected individuals in patch i is denoted by z_i (these are integer numbers in the Gillespie simulation). This way, the various terms in equations (1-2) describe the propensities of all the possible stochastic events. For example, $\mathcal{P}_1^{(i)} = dx_i$ is the propensity for a death event of an uninfected individual in patch i , and $\mathcal{P}_2^{(i)} = \beta x_i z_i$ is the propensity for an infection event in patch i where a single individual becomes infected. For patch i , there are also the following propensities:

$$\mathcal{P}_3^{(i)} = \left[rx_i \left(1 - \frac{x_i + z_i}{K} \right) \right]_+, \quad \mathcal{P}_4^{(i)} = az_i, \quad \mathcal{P}_5^{(i)} = \mu x_i, \quad \mathcal{P}_6^{(i)} = \mu z_i,$$

where $[\dots]_+$ stands for the positive part. Let us define the total patch propensity $\mathcal{P}^{(i)} = \sum_{k=1}^6 \mathcal{P}_k^{(i)}$ for $1 \leq i \leq \mathcal{N}$, and the total system propensity as $\mathcal{P} = \sum_{i=1}^{\mathcal{N}} \mathcal{P}^{(i)}$.

The algorithm proceeds as a sequence of steps. For each step, we first determine which patch is chosen for an updating event. This is done by choosing patch i with probability $\mathcal{P}^{(i)}/\mathcal{P}$ for each i . Once the updating patch is chosen, we need to determine what update happens. This is done by picking update k with probability $\mathcal{P}_k^{(i)}/\mathcal{P}^{(i)}$ for each $k \in \{1, \dots, 6\}$. Once the update is chosen, the number of individuals is changed accordingly. For example, if update $k = 2$ is chosen, then the number of individuals in patch

i changes as $x_i \rightarrow x_i - 1$ and $z_i \rightarrow z_i + 1$. Note that if a migration update is chosen (for example, $k = 5$), then the number of individuals in patch i decreases ($x_i \rightarrow x_i - 1$), and also the number of individuals increases in a randomly chosen patch m (with all patches being equally likely to be picked): $x_m \rightarrow x_m + 1$. Finally, we determine the time-increment, that is, the length of time, Δt , that elapsed between the previous update and the current update. This is done by setting Δt to be a random number exponentially distributed with the mean $1/\mathcal{P}$.

Trajectories obtained by using this algorithm are characterized by the mean values approximately given by equations (1-2).

To numerically study mutant fixation in the patch model, a procedure similar to the ABM was followed. Initially, a square block of patches was populated with a number of wild-type, uninfected individuals, and in the middle of that block, a smaller block of patches contained both infected and uninfected, wild-type individuals (the precise numbers used were 30 uninfected individuals per patch, and in addition 10 infected individuals per patch in the infected block). The Gillespie algorithm described above was followed until a quasi-equilibrium state was reached, at which point mutant(s) were introduced, as described in the main text.

1.2 A coarse-grained description

Let us define an infected patch as a patch that contains at least one infected cell (and it may or may not contain uninfected cells). An uninfected patch contains at least one uninfected cell and no infected cells. Instead of focusing on the dynamics of infected and uninfected cells within patches, we can consider the populations of infected and uninfected patches within the grid, see figure S1

Suppose the number of uninfected patches is X , and the number of infected patches is Z . Let us denote the mean number of cells in an uninfected patch as $w_x^{(X)}$; the mean number of infected cells in an infected patch as $w_z^{(Z)}$, and the mean number of uninfected cells in an infected patch as $w_x^{(Z)}$. We

Figure S1: A schematic of the coarse-grain model. (a) Patches can be uninfected, infected, or empty; here green and red dots denote uninfected and infected cells. (b) An infected patch can become uninfected after a time, T_{inf} . (c) An infected patch can become extinct after a time, T_{ext} . (d) The coarse-grained description includes these two processes and a process of colonization of empty patches. Here the blue arrows indicate which populations contribute to which rates.

have the following coarse-grained dynamics of patches:

$$\dot{X} = \mu(Xw_x^{(X)} + Zw_x^{(Z)}) \left(1 - \frac{X+Z}{\mathcal{N}}\right) P_{col} - \mu Zw_z^{(Z)} \frac{X}{\mathcal{N}} P_{inf}, \quad (3)$$

$$\dot{Z} = \mu Zw_z^{(Z)} \frac{X}{\mathcal{N}} P_{inf} - \frac{Z}{T_{ext}}. \quad (4)$$

Here, the first term in the right hand side of equation (3) is the rate at which all the uninfected cells in the system ($Xw_x^{(X)} + Zw_x^{(Z)}$) migrate and land on an empty patch (probability to hit an empty patch is $1 - (X + Z)/\mathcal{N}$), which is followed by a successful colonization (probability P_{col}). The second term in equation (3), as well as the first term in equation (4), describes the rate at which infected cells ($Xw_z^{(Z)}$) migrate to one of the uninfected patches (probability X/\mathcal{N}) and that infection takes off (probability P_{inf}). The last term in equation (4) is the rate at which infected patches go extinct, with T_{ext} denoting the mean extinction time of an infected patch. The probability of successful colonization, P_{col} , and the probability of successful infection

spread, P_{inf} , can be approximated as follows,

$$P_{col} = 1 - d/r, \quad (5)$$

$$P_{inf} = 1 - 1/R_0 = 1 - (w_x^{(X)}\beta/a)^{-1}. \quad (6)$$

Other parameters of system (4-4), such as the mean time of extinction, P_{ext} , and mean population sizes $w_x^{(X)}$, $w_z^{(Z)}$, and $w_x^{(Z)}$, are calculated from microscopic patch dynamics, as described next.

We will denote by $S(t)$ the number of uninfected cells and by $I(t)$ the number of infected cells in a typical patch.

Uninfected patches. We have, inside the uninfected patches:

$$\dot{S} = rS(1 - S/K) - dS - \mu S + \mu(Xw_x^{(X)} + Zw_x^{(Z)})\frac{1}{\mathcal{N}}, \quad (7)$$

$$S(0) = 1. \quad (8)$$

The life-span of an uninfected patch is defined by the timing of infection, T_{inf} . Infections happen at the rate $\frac{1}{\mathcal{N}}\mu Zw_z^{(Z)}P_{inf}$, so we have

$$T_{inf} = \frac{\mathcal{N}}{\mu Zw_z^{(Z)}P_{inf}}. \quad (9)$$

The mean population of an uninfected patch is therefore

$$w_x^{(X)} = \frac{1}{T_{inf}} \int_0^{T_{inf}} S(t) dt. \quad (10)$$

Infected patches. We have, inside the infected patches:

$$\dot{S} = rS(1 - (S + I)/K) - dS - \mu S + \mu(Xw_x^{(X)} + Zw_x^{(Z)})\frac{1}{\mathcal{N}} - \beta SI,$$

$$\dot{I} = \beta SI - aI - \mu I + \mu Zw_z^{(Z)}\frac{1}{\mathcal{N}},$$

$$S(0) = w_x^{(X)}, \quad (11)$$

$$I(0) = 1. \quad (12)$$

The time to extinction, $T_{ext} > 0$, is approximated by obtaining the time that corresponds to the first minimum of the function $I(t)$. The mean population

sizes are then given by

$$w_z^{(Z)} = \frac{1}{T_{ext}} \int_0^{T_{ext}} I(t) dt, \quad (13)$$

$$w_x^{(Z)} = \frac{1}{T_{ext}} \int_0^{T_{ext}} S(t) dt. \quad (14)$$

Coarse-grained method numerical procedure. The goal is, given all the system parameters, to predict the expected (infected and uninfected) patch numbers and expected number of (infected and uninfected) individuals inside patches. Solutions (X, Z) of system (3-4), as well as values $w_x^{(X)}$, $w_z^{(Z)}$, and $w_x^{(Z)}$ were used to approximate these values. The following steps were implemented:

- 0) Use an initial guess for the 5 quantities, $(X, Z, w_x^{(X)}, w_z^{(Z)}, w_x^{(Z)})$.
- 1) Evaluate parameters P_{inf} and T_{inf} , equations (6) and (9).
- 2) Numerically solve the system for infected patches (equations (11-12)) up to some large value of time, T_{max} .
- 3) Obtain $T_{ext} > 0$ by locating the first minimum of the function $I(t)$ in $(0, T_{max})$.
- 4) Numerically solve the system for uninfected patches (equations (7-8)) up to the value T_{ext} .
- 5) Evaluate the updated values of $w_x^{(X)}$ using the solution in 4), equation (10); evaluate the updated values of $w_z^{(Z)}$ and $w_x^{(Z)}$ using the solution in 2), equations (13-14).
- 6) Perform an iteration of a discretized version of equations (3-4) to find updated values of X, Z :

$$\begin{aligned} \dot{X}^{new} = & X^{old} + \Delta t \left(\mu(X^{old} w_x^{(X)} + Z^{old} w_x^{(Z)}) \left(1 - \frac{X^{old} + Z}{\mathcal{N}} \right) P_{col} \right. \\ & \left. - \mu Z^{old} w_z^{(Z)} \frac{X^{old}}{\mathcal{N}} P_{inf} \right), \end{aligned} \quad (15)$$

$$\dot{Z}^{new} = Z^{old} + \Delta t \left(\mu Z^{old} w_z^{(Z)} \frac{X^{old}}{\mathcal{N}} P_{inf} - \frac{Z^{old}}{T_{ext}} \right). \quad (16)$$

7) Repeat steps (0-6), until convergence is reached.

1.3 Stochastic simulations and coarse-grained predictions

Stochastic Gillespie simulations of system (1-2) were performed. The simulations were initiated by placing a relatively small block of occupied patches in the middle of a the grid, with the rest of the patches initially empty. All the occupied patches were uninfected except for a smaller block in the middle. For example, in a 40×40 grid, we started with 7×7 occupied patches in the middle, of which the central 3×3 block was infected. The initial populations of the occupied patches were just below the carrying capacity, and the initial population of infected patches contained a third of infected cells. Simulations of this type were run for a range of migration rate values, μ . Figure S2 show typical dynamics inside a single patch in a multi-patch simulation, under a lower (panel (a)) and a higher (panel (b)) migration rate.

Figure S2: Typical patch trajectories under (a) low migration ($\mu = 0.05$) and (b) high migration ($\mu = 0.15$). The numbers of uninfected (blue) and infected (yellow) individuals are plotted as a function of time for a single patch. The rest of the parameters are: $r = 0.6, d = 0.1, \beta = 1.5, a = 0.5, K = 20, \mathcal{N} = 400$.

For each value of the migration rate, we obtained numerical counts of infected and uninfected patch numbers, as well as the numbers of individuals inside patches, at quasi-equilibrium. To count patch numbers, we assumed that a patch is uninfected if it contains at least one uninfected individual and

no infected individuals; a patch is infected if it contains at least one infected individual. Figure S3 shows a comparison of stochastic simulations (dots with vertical bars, representing ensemble means and standard deviations) with the coarse-grained method predictions (dashed lines), for patch numbers (X, Z , see panel (a)) and the numbers of individuals ($w_x^{(X)}, w_z^{(Z)}, w_x^{(Z)}$, see panel (b)).

Figure S3: Comparison of stochastic simulations (dots with vertical bars, showing means and standard deviations), and the coarse-grained method predictions (dashed lines). (a) The numbers of uninfected (blue) and infected (yellow) patches are plotted for different values of the migration rate, μ . (b) The numbers of uninfected individuals in uninfected patches ($w_x^{(X)}$, blue), infected individuals ($w_z^{(Z)}$, yellow), and uninfected individuals in infected patches ($w_x^{(Z)}$, green) are plotted for different values of μ . The rest of the parameters are as in figure S2.

2 A coarse-grained approach in the presence of mutants

2.1 The patch model in the presence of two types of hosts

In the presence of two types of mutants, the dynamics in the patch system are described by the following ODEs,

$$\dot{x}_i = r_x x_i \left(1 - \frac{x_i + y_i + z_i}{K}\right) - d_x x_i - \beta x_i z_i + \mu \left(\frac{\sum_{k=1}^{\mathcal{N}} x_k}{\mathcal{N}} - x_i\right), \quad (17)$$

$$\dot{y}_i = r_y y_i \left(1 - \frac{x_i + y_i + z_i}{K}\right) - d_y y_i - \beta y_i z_i + \mu \left(\frac{\sum_{k=1}^{\mathcal{N}} y_k}{\mathcal{N}} - y_i\right), \quad (18)$$

$$\dot{z}_i = \beta(x_i + y_i)z_i - a z_i + \mu \left(\frac{\sum_{k=1}^{\mathcal{N}} z_k}{\mathcal{N}} - z_i\right), \quad 1 \leq i \leq \mathcal{N}, \quad (19)$$

where x_i, y_i , and z_i are respectively the number of uninfected wild-type cells, uninfected mutant cells, and infected cells. The division rates of wild-type and mutant cells are given by r_x and r_y respectively. The death rates of wild-type and mutant cells are given by d_x and d_y respectively. It is assumed that while mutants may differ from the wild type cells by their division or death rates, both types of host are equally susceptible to infection (they are characterized by the same infectivity, β), they migrate with the same rate μ , and the death rate of infected cells, a , does not depend on their type. This is a generalization of system (1-2), and was also simulated by using the Gillespie algorithm.

2.2 A coarse-grained system in the presence of two types of hosts

Let us extend the coarse-grained approach, equations (3-4), to systems with two types of host, wild type and mutant cells. In order for this approach to work, most nonempty patches in the system should contain either wild-type or mutant individuals. If a significant fraction of patches contains both types, the approach is not valid (see below for the applicability conditions). Denote by X and Y the numbers of uninfected wild-type and mutant patches,

respectively. The numbers of wild-type and mutant infected patches are denoted respectively by Z_x and Z_y . The coarse-grained system of equations in this case is given by

$$\begin{aligned}\dot{X} &= -\mu(Yw_y^{(Y)} + Z_yw_y^{(Z_y)})\frac{X}{\mathcal{N}}P_{x\rightarrow y} + \mu(Xw_x^{(X)} + Z_xw_x^{(Z_x)})\frac{Y}{\mathcal{N}}P_{y\rightarrow x} \\ &+ \mu(Xw_x^{(X)} + Z_xw_x^{(Z_x)})\left(1 - \frac{X+Y+Z_x+Z_y}{\mathcal{N}}\right)P_{col}^x - \mu(Z_xw_z^{(Z_x)} + Z_yw_z^{(Z_y)})\frac{X}{\mathcal{N}}P_{inf}^x, \quad (20)\end{aligned}$$

$$\begin{aligned}\dot{Y} &= \mu(Yw_y^{(Y)} + Z_yw_y^{(Z_y)})\frac{X}{\mathcal{N}}P_{x\rightarrow y} - \mu(Xw_x^{(X)} + Z_xw_x^{(Z_x)})\frac{Y}{\mathcal{N}}P_{y\rightarrow x} \\ &+ \mu(Yw_y^{(Y)} + Z_yw_y^{(Z_y)})\left(1 - \frac{X+Y+Z_x+Z_y}{\mathcal{N}}\right)P_{col}^y - \mu(Z_xw_z^{(Z_x)} + Z_yw_z^{(Z_y)})\frac{Y}{\mathcal{N}}P_{inf}^y, \quad (21)\end{aligned}$$

$$\dot{Z}_x = \mu(Z_xw_z^{(Z_x)} + Z_yw_z^{(Z_y)})\frac{X}{\mathcal{N}}P_{inf}^x - \frac{Z_x}{T_{ext}^x}, \quad (22)$$

$$\dot{Z}_y = \mu(Z_xw_z^{(Z_x)} + Z_yw_z^{(Z_y)})\frac{Y}{\mathcal{N}}P_{inf}^y - \frac{Z_y}{T_{ext}^y}. \quad (23)$$

As before, the populations of individuals in different patches are denoted with w , where the subscript refers to the population type (x/y for uninfected w.t./mutants, and z for infected cells), and the superscript refers to the patch type.

Compared to system (3-4), the equations in the presence of mutants have two new terms. The first term in the right hand side of equations (20) and (21) describes the rate at which uninfected mutant cells (totaling $Yw_y^{(Y)} + Z_yw_y^{(Z_y)}$) migrate to an uninfected wild-type patch (probability X/\mathcal{N}), and subsequently, the patch is ‘‘converted’’ to becoming a mutant patch by means of stochastic fixation of mutants (probability $P_{x\rightarrow y}$). Similarly, the second term in the right hand side of equations (20) and (21) describes the conversion of mutant patches to wild-type patches. The probability of fixation of mutants in a patch of wild-type individuals, and also that for wild-type individuals in a patch of mutants, are given by

$$P_{x\rightarrow y} = \frac{1 - 1/(1+s)}{1 - 1/(1+s)^{w_x^{(X)}}}, \quad P_{y\rightarrow x} = \frac{1 - (1+s)}{1 - (1+s)^{w_y^{(Y)}}}, \quad s = \frac{r_y d_x}{d_y r_x} - 1.$$

Here, s is the selection coefficient and $w_x^{(X)}$ and $w_y^{(Y)}$ are patch sizes for the two types of cells. The above formulas are the Moran process approximations for the fixation probabilities for populations with demographic fluctuations [1].

In addition, in the presence of two types of hosts, some of the rates now have a superscript that specifies the type of a patch where the process takes place. For example, P_{col}^x and P_{col}^y denote the probability of successful colonization of an empty patch by wild type and mutant cells, respectively, and are given by $P_{col}^x = 1 - d_x/r_x$, $P_{col}^y = 1 - d_y/r_y$.

2.3 Quantification of selection dynamics

The total number of uninfected wild-type cells in system (20-23) is given by $Xw_x^{(X)} + Z_xw_x^{(Z_x)}$, and the total number of uninfected mutants by $Yw_y^{(Y)} + Z_yw_y^{(Z_y)}$. The two contributions are unequal, with the large majority of uninfected cells coming from uninfected patches. Therefore, we can ignore¹ the terms $Z_xw_x^{(Z_x)}$, $Z_yw_y^{(Z_y)}$ compared to $Xw_x^{(X)}$, $Yw_y^{(Y)}$. The resulting 3-equation system is presented in the main text, where we denoted $Z = Z_x + Z_y$.

We would like to determine the selection pressure experienced by the mutant patches, given that they are at low numbers compared to the number of wild-type patches. To proceed, we will find the equilibrium solution of system (20-23) in the absence of mutants ($X = X_{eq}$, $Y = 0$, $Z_x = Z_{eq}$, $Z_y = 0$), and then linearize the system around this solution, to obtain the equation for \dot{Y} .

We have the following equilibrium solution in the absence of mutants:

$$X_{eq} = \frac{\mathcal{N}}{\mu P_{inf}^x T_{ext} w_z^{(Z)}}, \quad Z_{eq} = \frac{\mathcal{N} w_x^{(X)} P_{col}^x}{\mu w_z^{(Z)} P_{inf}^x T_{ext}} \frac{\mu w_z^{(Z)} P_{inf}^x T_{ext} - 1}{w_x^{(X)} P_{col}^x + w_z^{(Z)} P_{inf}^x}.$$

Linearizing equation (21), we obtain

$$\dot{Y} = (\nu_{col} + \nu_{x \rightarrow y} - \nu_{inf} - \nu_{y \rightarrow x})Y, \quad (24)$$

where the four rates on the right hand side correspond to the four processes that can change the number of mutant patches:

¹Note that this step significantly simplifies all the expressions, but it is not necessary for the method to work. If we keep contributions $Z_xw_x^{(Z_x)}$, $Z_yw_y^{(Z_y)}$, then the equilibrium value Z_{eq} can be obtained from a quadratic, rather than linear, equation, and the system for mutants will contain linear ODEs for \dot{Y} and \dot{Z}_y , where the latter equation can be solved in quasi-equilibrium, reducing the system to a single equation for \dot{Y} .

- Colonization of empty patches by mutant cells, which increases the number of mutant patches with the per-patch rate

$$\nu_{col} = \mu w_y^{(Y)} \left(1 - \frac{X_{eq} + Z_{eq}}{\mathcal{N}} \right) P_{col}^y,$$

- Conversion of wild-type patches to mutant patches, which also increases the number of mutant patches, with the per-patch rate

$$\nu_{x \rightarrow y} = \mu w_y^{(Y)} \frac{X_{eq}}{\mathcal{N}} P_{x \rightarrow y},$$

- Infection of mutant patches which leads to their subsequent extinction (and decreases their number) at the per-patch rate

$$\nu_{inf} = \mu Z_{eq} w_z^{(Z)} \frac{1}{\mathcal{N}} P_{inf}^y,$$

- Conversion of mutant patches to wild-type patches, which decreases the number of mutant patches with the per-patch rate

$$\nu_{y \rightarrow x} = \mu X_{eq} w_x^{(X)} \frac{1}{\mathcal{N}} P_{y \rightarrow x}.$$

To better understand the role of these processes in patch selection, we can envisage a corresponding coarse-grained stochastic process, where the number of mutant patches is denoted as $j \in [0, N]$, and during an infinitesimal time, Δt , the following processes take place:

- The number of mutant patches can increase by one with probability $P(j \rightarrow j+1) = j(\nu_{col} + \nu_{x \rightarrow y})\Delta t$;
- The number of mutant patches can decrease by one with probability $P(j \rightarrow j-1) = j(\nu_{inf} + \nu_{y \rightarrow x})\Delta t$;
- The number of mutant patches can remain constant with probability $P(j \rightarrow j) = 1 - (P(j \rightarrow j+1) + P(j \rightarrow j-1))$.

Denote by ρ_j the probability that starting from j mutant patches, the system reaches the state where all patches are mutant ($j = N$). We have $\rho_0 = 0$, $\rho_N = 1$, and

$$\rho_j = P(j \rightarrow j+1)\rho_{j+1} + P(j \rightarrow j-1)\rho_{j-1} + \rho_j(1 - (P(j \rightarrow j+1) + P(j \rightarrow j-1))), \quad 0 < j < N$$

which is equivalent to

$$\rho_{j+1}(\nu_{col} + \nu_{x \rightarrow y}) + \rho_{j-1}(\nu_{inf} + \nu_{y \rightarrow x}) - \rho_j(\nu_{col} + \nu_{inf} + \nu_{x \rightarrow y} + \nu_{y \rightarrow x}) = 0.$$

The probability of mutant fixation can be obtained by the usual methods (see e.g. [2]) and is given by

$$\rho_i = \frac{1 - 1/(1 + s_p)^i}{1 - 1/(1 + s_p)^N},$$

where $N \approx X_{eq}$ and the quantity s_p can be interpreted as the effective patch selection coefficient. It is given by

$$s_p = \frac{\nu_{col} + \nu_{x \rightarrow y}}{\nu_{inf} + \nu_{y \rightarrow x}} - 1. \quad (25)$$

This expression for the effective patch selection coefficient can be rewritten in a more intuitive way:

$$s_p = \frac{R_{ext-rec} s_p^{ext-rec} + R_{conv} s_p^{conv}}{R_{ext-rec} + R_{conv}}, \quad (26)$$

where the rates of the two processes are

$$R_{conv} = \frac{X_{eq}}{\mathcal{N}} \mu w_x^{(X)} P_{y \rightarrow x} = \frac{w_x^{(X)} P_{y \rightarrow x}}{w_z^{(Z)} P_{inf}^x T_{ext}}, \quad (27)$$

$$R_{ext-rec} = \frac{Z_{eq}}{\mathcal{N}} \mu w_z^{(Z)} P_{inf}^y = \frac{P_{inf}^y w_x^{(X)} P_{col}^x (\mu w_z^{(Z)} P_{inf}^x T_{ext} - 1)}{P_{inf}^x T_{ext} (w_x^{(X)} P_{col}^x + w_z^{(Z)} P_{inf}^x)}, \quad (28)$$

and the associated effective patch selection coefficients are

$$s_p^{conv} = \frac{w_y^{(Y)} P_{x \rightarrow y}}{w_x^{(X)} P_{y \rightarrow x}} - 1, \quad s_p^{ext-rec} = \frac{w_y^{(Y)} P_{col}^y P_{inf}^x}{w_x^{(X)} P_{col}^x P_{inf}^y} - 1. \quad (29)$$

Note that both expressions (27) and (28) are “out” rates, as they stand for the conversion out of Y and extinction of Y . In the case where the quantities in (29) are small compared to 1, they can be replaced by the “in” rates, and the difference will be small in s_p^{conv} and $s_p^{ext-rec}$.

The regime where the extinction-recolonization process is dominant is characterized by

$$Z_{eq}w_z^{(Z)}P_{inf}^y \gg X_{eq}w_x^{(X)}P_{y \rightarrow x},$$

or, denoting by N_{inf} and N_{uninf} the total expected number of infected and uninfected cells at the mutant-free equilibrium, we have the condition

$$N_{inf}P_{inf}^y \gg N_{uninf}P_{y \rightarrow x}.$$

If the process of extinction-recolonization is predominant, then the probability of mutant fixation, starting from a single fully mutant deme, is given by

$$\rho_{patch} = \frac{1 - 1/(1 + s_e^{ext-rec})}{1 - 1/(1 + s_e^{ext-rec})X_{eq}}. \quad (30)$$

2.4 Comparison with simulations

If the process of conversion is dominant, then we expect the probability of mutant fixation to be similar to that of the Moran process. On the other hand, if the extinction-recolonization process is predominant compared to the conversion process, we expect selection to be significantly weaker and the probability of fixation much larger (smaller) for disadvantageous (advantageous) mutants.

To test this theory, it is instructive to use the parameter regime where the process of extinction-recolonization process is predominant. This is observed when the mutant and wild type cells are typically separated, that is, they do not co-occur in the same demes. Such separation happens for example if many demes are unoccupied (allowing for recolonization by a single type), and when demes' life-span is relatively short (due to intense infection), which precludes accumulation of cells of the different types due to migration.

While spatial separation of mutants and wild-types is typical in simulation with spatially restricted migration, it is more difficult to achieve under non-spatial migration. To increase the degree of separation of mutants and wild types (and thus to amplify the contribution of extinction-recolonization process in the absence of space), we assumed that infected cells migrated at a faster rate than uninfected cells. As a result, demes' lifespans became shorter (due to intense infection) and conversion did not play a significant role.

Figure S4 shows a typical result of a Gillespie simulation of a non-spatial system where mutant and wild-type cells are largely separated (for a detailed

Figure S4: Dynamics of a patch system with non-spatial migration. (a) The total number of uninfected wild type (blue) and mutant (orange) individuals, as well as infected (green) individuals, as a function of frame number. (b) The number of uninfected (green), infected (red), and empty (gray) patches, as a function of frame number. (c) The fraction of uninfected patches (that is, patches that only contain either mutant or wild type cells), blue, and mixed (both mutant and wild type cells, orange) patches, as a function of frame number. (d) A histogram showing the number of uninfected nonempty patches characterized by different fractions of mutant cells – a congregate over all 500 frames. The parameters are $r_x = 0.7, r_y = 1.02r_x, d_x = d_y = 0.6, K = 200, a = 0.1, \beta = 0.04, N = 1600$. The migration rate is $\mu = 0.01$ for uninfected individuals and twice that amount for infected individuals.

description of the type of Gillespie algorithms used here, see Section 1.1). Mutant individuals were assumed to have a 2% advantage in the division rate. In a simulation run where mutant individuals expand from low numbers to reach a majority, we recorded the state of the patches **once the fraction of mutants reached 20%**. Panel (a) shows a time-series for the total (over all patches) numbers of uninfected wild type, uninfected mutants, and infected individuals, to show that the overall number of mutants varies from low to high numbers. Panel (b) shows three types of patches: uninfected (that is, patches that only contain uninfected individual, regardless of their type), infected (that is, patches that contain at least one infected individual), and empty. In particular, we observe that the number of empty patches is relatively high (the total number of patches is 1600). Panel (c) plots time-series

that characterize the degree of mixing of wild type and mutant individuals: for all uninfected nonempty patches, we plot the fraction of patches that either contain mutant or wild type cells (the homogeneous patches, blue); similarly, in orange, we plot the fraction of mixed patches (that is uninfected patches that contain at least one wild-type and at least one mutant individual). We observe that the fraction of mixed patches stays low throughout the frames. Finally, panel (d) shows a histogram of mutant fraction in the uninfected patches. **It represents the mutant fractions collected in the course of the simulation, sampled 500 times (after every 10,000 Gillespie updates, approximately 1 time unit) during the time-interval depicted in panels (a-c).** We can see again that the majority of patches are either 100% mutant or 100% wild-type.

Note that most simulations that start with a low number of mutants will result in mutant extinction without ever seeing a significant number of mutants; for such runs, it is not surprising that most of the patches are homogeneous (and contain wild-type cells). The simulation shown in figure S4 was chosen such that mutants do rise to a significant percentage. It demonstrates that even in such cases, the two types of individuals remain mostly separated (that is, rarely inhabit the same patch).

Figure S5: The coarse-grained approach applied to the system in figure S4. Iterations of steps (0-6) of section 1.2 are shown for the deme numbers (a) and population sizes (b). The means and standard deviations obtained by stochastic simulations are shown on the right side of each graph. Parameters are as in figure S4.

For this system, we demonstrate that the method described above pro-

duces a very good prediction of the mutant fixation probability. Figure S5 shows the result of iterations of system (15-16), to obtain an approximation of the deme numbers and population sizes. The means and standard deviations obtained by stochastic simulations are shown on the right side of each graph.

Figure S6: Mutant fixation probability as a function of mutant selection coefficient, s . The blue line is the coarse-grained prediction, equation (30). The red line is the Moran prediction, equation (31). The points with small vertical bars represent the mean and standard deviation of the probability of fixation determined by stochastic simulations for three different values of s . The rest of the parameters are as in figure S4.

The probability of mutant fixation, starting with a single fully-mutant deme, is given by equation (30). Figure S6 compares this calculation (blue line) with the result for the usual Moran prediction,

$$\rho_{Moran} = \frac{1 - 1/(1 + s)^{w_x^{(X)}}}{1 - 1/(1 + s)^{w_x^{(X)} X_{eq}}}, \quad (31)$$

which is shown by the red line. Numerical simulations of the stochastic system were performed to estimate the probability of mutant fixation; the results are shown by points with small standard deviations that correspond to three different values of the mutant selection coefficient s . In these simulations, first, a patch system in the absence of mutants was allowed to reach a quasi-equilibrium. Then, at a time when the number of uninfected cells was given

by N_u (the numerically determined mean number at quasi-equilibrium), a patch that contained at least one uninfected cell was selected randomly, and all uninfected cells were replaced by mutant uninfected cells. The simulation was stopped when either the mutants reached fixation or were extinct. This process was repeated 15,480 times, for each value of s , yielding the fraction of the runs that resulted in mutant fixation.

We observe that the prediction of the coarse-grained approach is very similar to the observed probability of mutant fixation, and that the Moran prediction corresponds to a much higher fixation probability for advantageous mutants.

References

- [1] Todd L Parsons and Christopher Quince. Fixation in haploid populations exhibiting density dependence i: the non-neutral case. *Theoretical population biology*, 72(1):121–135, 2007.
- [2] Samuel Karlin and Howard E Taylor. *A second course in stochastic processes*. Elsevier, 1981.

Reviewers' Comments:

Reviewer #1:

None

Reviewer #2:

Remarks to the Author:

The authors have carefully addressed all my comments and I still think this is a very good paper (even before these changes were implemented). However, there is one point where I continue to politely disagree, the authors write in the main text "stochastic Gillespie simulations of ODEs". In my thinking, there are many possible Gillespie simulations that are described by a single ODE, so the authors pick one. However, in the new Section 1.1 of the SI they clearly describe how they do it and that each of the separate terms is a different process. Thus, it is mostly the formulation "stochastic Gillespie simulations of ODEs" that I find slightly unlucky.

Another minor issue: The algorithm described is using a random number to determine what happens (chosen according to P_i/P) and a second one to determine the time. According to my knowledge, the advance of Gillespie (1976) was to choose numbers that do both at the same time. He suggested to use exponentially distributed numbers for each reaction and implement the reaction that is fastest only. I think this is what most people call Gillespie algorithm, but the term may be used differently in different fields.

Otherwise, I am looking forward to see this published.

Mutant fixation in the presence of a natural enemy

Reply to the referee report

We would like to thank the referees and the editor for reading the paper, and for their positive assessments of our revisions. Below please find a point-by-point reply, where the remaining comments of Referee 2 are addressed. We also edited the manuscript to comply with the journal requirements that were given in the checklist. Below this reply, we append versions of the manuscript and the Supplementary Notes that highlight the changes.

Reviewer #2 (Remarks to the Author):

The authors have carefully addressed all my comments and I still think this is a very good paper (even before these changes were implemented).

We thank the referee for this positive assessment of the paper.

However, there is one point where I continue to politely disagree, the authors write in the main text “stochastic Gillespie simulations of ODEs”. In my thinking, there are many possible Gillespie simulations that are described by a single ODE, so the authors pick one. However, in the new Section 1.1 of the SI they clearly describe how they do it and that each of the separate terms is a different process. Thus, it is mostly the formulation “stochastic Gillespie simulations of ODEs” that I find slightly unlucky.

We agree that there are many possible Gillespie-type simulations whose average behavior is approximated by a given system of ODEs, see e.g. a review in [Ramaswamy, R., González-Segredo, N., & Sbalzarini, I. F. (2009). *The Journal of chemical physics*, 130(24)]. To avoid confusion, in the new version of the manuscript we referred to our algorithm as a Gillespie-type stochastic simulation algorithm (SSA). For example, to introduce the algorithm in Section 1.1 of the Supplement, we now write: “*Here we will describe the Gillespie-type stochastic simulation algorithm that we used to sample trajectories from the mean-field description, by using the example of system (1-2). Our algorithm is one of many possible stochastic methodologies, see e.g. [Ramaswamy et al (2009)] for a review.*”

Another minor issue: The algorithm described is using a random number to determine what happens (chosen according to P_i/P) and a second one to determine the time. According to my knowledge, the advance of Gillespie (1976) was to choose numbers that do both at the same time. He suggested to use exponentially distributed numbers for each reaction and implement the reaction that is fastest only. I think this is what most people call Gillespie algorithm, but the term may be used differently in different fields.

Our algorithm uses two random numbers for each step: one to determine which “reaction” will happen and the second one to determine the time-interval that has elapsed. This is the same method as referred to as a direct method (DM), which is described concisely, for example, in Appendix A of [Ramaswamy, R., González-Segredo, N., & Sbalzarini, I. F. (2009). *The Journal of chemical physics*, 130(24)]. In the new version of the manuscript, we have added the exact reference to explain this choice.